# BATCH SPECULATIVE DECODING DONE RIGHT

## ABSTRACT

Speculative decoding speeds up LLM inference by using a small draft model to propose multiple tokens that a target model verifies in parallel. Extending this idea to batches is essential for production serving, but it introduces the ragged tensor problem: sequences in the same batch accept different numbers of draft tokens, breaking right-alignment and corrupting position IDs, attention masks, and KV-cache state. We show that several existing batch implementations violate output equivalence—the fundamental requirement that speculative decoding must produce identical token sequences to standard autoregressive generation. These violations occur precisely due to improper handling the ragged tensor problem. In response, we (1) characterize the synchronization requirements that guarantee correctness, (2) present a correctness-first batch speculative decoding EQSPEC that exposes realignment as consuming 40% of overhead, and (3) introduce EXSPEC, which maintains a sliding pool of sequences and dynamically forms same-length groups, to reduce the realignment overhead while preserving per-sequence speculative speedups. On the SpecBench dataset, across Vicuna-7B/68M, Qwen3-8B/0.6B, and GLM-4-9B/0.6B target/draft pairs, our approach achieves up to 3× throughput improvement at batch size 8 compared to batch size 1, with efficient scaling through batch size 8, while maintaining 95% output equivalence. Our method requires no custom kernels and integrates cleanly with existing inference stacks.

## 1 INTRODUCTION

Speculative decoding (Leviathan et al., 2023; Chen et al., 2023) accelerates LLM inference by using a small draft model to propose multiple tokens that the target model verifies in parallel, shifting from memory-bound sequential generation to compute-intensive verification and delivering single-sequence speedups (Xia et al., 2024). Batch speculative decoding aims to combine this per-sequence acceleration with standard batching by processing multiple sequences (batch dimension) while verifying multiple draft tokens per sequence (sequence dimension). The core challenge is the ragged-tensor effect (Qian et al., 2024): in each verification round, sequences accept a single series of the proposed draft tokens, but prefix lengths differ across sequences (e.g., one accepts five tokens while another accepts one), misaligning sequence lengths. This raggedness violates the rectangular-tensor assumption required for GPU-parallel execution.

Figure 1 highlights a critical, often overlooked issue in batch speculative decoding: methods with impressive throughput can produce **corrupted** outputs. For lossless acceleration, speculative decoding must yield **identical** outputs to standard autoregressive generation (Leviathan et al., 2023). As a reference point, the widely used HuggingFace implementation (Wolf et al., 2020) preserves this guarantee, but only for batch size 1. By contrast, in our tests of public batch implementations—specifically, BSP (Su et al., 2023) and DSD (Yan et al., 2025)—this requirement is violated at batch sizes > 1, manifesting as repetitive tokens or <unk> symbols under greedy decoding rather than matching standard generation.

These failures share a single root cause—broken synchronization invariants (position tracking, attention, KV-cache) across ragged tensors. We first propose EQSPEC by formalizing these invariants and enforcing them with synchronization-aware scheduling (Section 3.1). Concretely, EQSPEC specifies the minimal synchronization needed for correctness and shows that realignment accounts for 40% of computation—an inherent cost of maintaining invariants across ragged tensors. This fundamental overhead helps explain why even production systems struggle: vLLM's speculative decoding under-

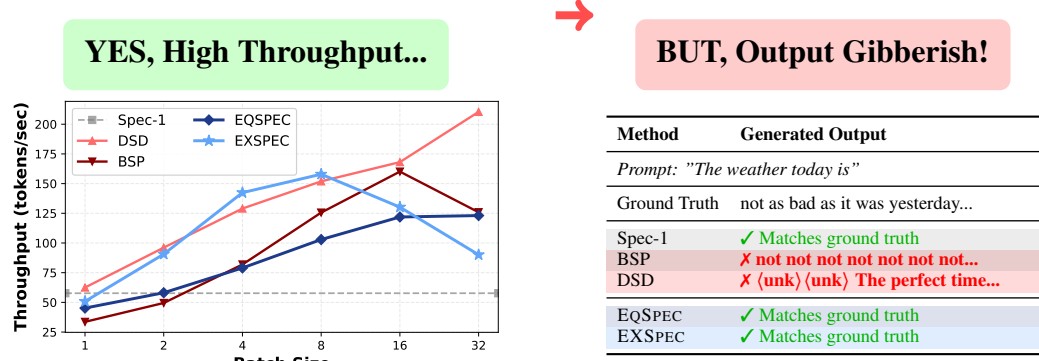

Figure 1: Batch speculative decoding on Vicuna-7B/68M: Existing methods achieve high throughput but **violate the fundamental requirement of output equivalence** by producing corrupted outputs. Our approach maintains perfect correctness while still achieving competitive performance.

performs its non-speculative baseline at higher batch sizes (leading to deprecation in its v1 engine), while SGLang with EAGLE (Li et al., 2024a) consistently exhibits negative speedups. Our analysis indicates that the superlinear growth of synchronization overhead is an inevitable cost of correctness in batch speculative decoding—a barrier no existing system has overcome.

To address this in practice, our main algorithm EXSPEC (Section 3.3) expands the scheduling scope: it maintains a sliding window of active sequences and dynamically groups those with identical lengths, eliminating realignment for homogeneous groups. This strategy preserves the scaling efficiency of standard batching—where GPU parallelism drives throughput—while retaining the per-sequence acceleration benefits of speculation. At batch size 8, we achieve a $3\times$ throughput improvement over batch size 1. Nevertheless, beyond batch size 8, throughput degrades as grouping success rates decline, forcing more frequent fallbacks to expensive realignment. Section 4 examines these scaling dynamics and the relationship between sequence diversity, grouping effectiveness, and alignment overhead.

Our experiments on SpecBench (Xia et al., 2024) yield two main results. First, **Correctness**: unlike prior approaches such as BSP and DSD, which suffer severe output corruption, our method preserves approximately 95% output equivalence across Vicuna-7B/68M (Zheng et al., 2023), Qwen3-8B/0.6B (Yang et al., 2025), and GLM-4-9B/0.6B (GLM et al., 2024) model pairs. Second, **Scalability**: at batch size 8, EXSPEC achieves up to a $3\times$ speedup over batch size 1. Our contributions are summarized as follows:

- We provide a correctness-first analysis of the ragged-tensor problem in batch speculative decoding, identifying precise synchronization requirements for correctness and explaining why existing methods fail (Section 2).

- We present a unified solution that maintains correctness through precise synchronization invariants while avoiding their overhead via cross-batch scheduling of same-length sequence groups (Section 3).

- We experimentally demonstrate that our approach achieves both $>95\%$ output correctness and positive scaling through batch size 8, successfully multiplying batch parallelism with per-sequence speculation gains, whereas production systems (vLLM, SGLang) exhibit negative speedups (Section 4).

## 2 DESIGN SPACE ANALYSIS

When sequences within a batch accept different numbers of draft tokens during verification, tensors become irregular, violating GPUs' requirement for rectangular layouts—this is the ragged-tensor problem illustrated in Figure 2. Despite batching's centrality to production deployments, existing implementations lack a principled design that preserves output equivalence with standard decoding while scaling with batch size. We identify three approaches to handle raggedness: *Masking*, *Rollback*, and *Dynamic Padding*. Yet, as our systematic analysis shows, current instantiations of these approaches fail to simultaneously maintain correctness and performance at scale. To close this

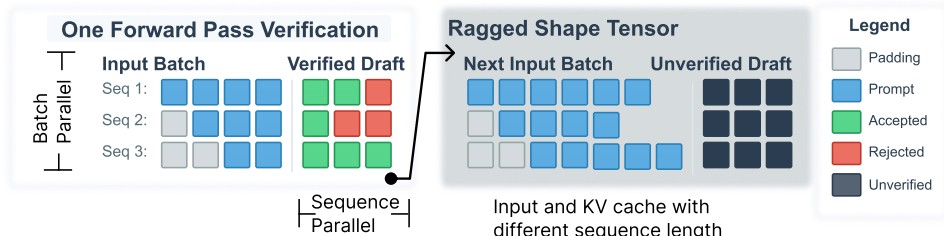

Figure 2: The ragged tensor problem in batch speculative decoding. Differing numbers of accepted draft tokens across sequences in the same batch lead to ragged-shaped input IDs tensors and KV Cache that disrupt subsequent batch operations.

**Algorithm 1** BatchVerify: Single Forward Pass

**Require:** Target model $\mathcal{M}_t$, Sequences $\mathcal{S}$, Draft tokens $D$, KV cache
**Ensure:** Accepted tokens $A$, Bonus tokens $B$
1: **if** first iteration **then**
2:     $\mathcal{X} \leftarrow \mathcal{S} \oplus D$
3: **else**
4:     $\mathcal{X} \leftarrow D$
5: $logits, KVCache \leftarrow \mathcal{M}_t(\mathcal{X}, KVCache)$
6: $pred\_tokens \leftarrow \arg\max(logits, dim=vocab)$
7:         ▷ *Vectorized first mismatch detection*
8: $matches \leftarrow (pred\_tokens = D)$
9: $J \leftarrow \arg\max(\neg matches, dim=seq)$
10:     ▷ *Ragged shape acceptance, no vectorization*
11: **for** each sequence $i$ in batch **do**
12:     $A[i] \leftarrow D[i][:j]$
13:         ▷ *Get bonus token from first mismatch*
14:     $bonus\_logit \leftarrow logits[i, |\mathcal{S}[i]| + j]$
15:     $B[i] \leftarrow \arg\max(bonus\_logit)$
16: **return** $A, B, KVCache$

**Algorithm 2** EQSPEC

**Require:** Draft model $\mathcal{M}_d$, Target model $\mathcal{M}_t$, Prompts $\mathcal{P}$, Max tokens $T$, Draft length $K$
**Ensure:** Generated sequences $\mathcal{S}$
1: $\mathcal{S} \leftarrow \text{Tokenize}(\mathcal{P})$     ▷ *Batch left padding*
2: $KVCache \leftarrow \emptyset$
3: **while** until max new tokens **do**
4:     **Phase 1: Draft Generation**
5:     $D \leftarrow \mathcal{M}_d.\text{Generate}(\mathcal{S}, K)$
6:     **Phase 2: Batch Verification**
7:     $A, B, KVCache \leftarrow$ $\text{BatchVerify}(\mathcal{M}_t, \mathcal{S}, D, KVCache)$
8:     ▷ *See Figure 3 for illustration on index offset.*
9:     **Phase 3: Unpad-Append-Repad**
10:     **for** each sequence $i$ in batch **do**
11:        $\mathcal{S}[i] \leftarrow \text{Unpad}(\mathcal{S}[i])$
12:        $\mathcal{S}[i] \leftarrow \mathcal{S}[i] \oplus A[i] \oplus B[i]$
13:     $\mathcal{S}, offset \leftarrow \text{BatchRepad}(\mathcal{S})$
14:     $KVCache \leftarrow \text{Realign}(KVCache, offset)$
15: **return** $\mathcal{S}$

gap, we first analyze the pitfalls of each approach and then introduce a correctness-first algorithmic design with explicit synchronization requirements for reliable batch speculative decoding.

✗ **Masking Approach (non-contiguous position IDs).** This approach operates directly on ragged tensors by masking rejected tokens in attention and reassigning position IDs so new tokens align with their content positions. Across verification rounds with varying rejections, sequences accumulate padding in various positions (middle and right), forming non-contiguous position IDs that standard Transformer implementations handle poorly. BSP (Su et al., 2023) attempts this via masking but fails to maintain position-ID consistency across iterations, yielding corrupted outputs (Figure 1). EAGLE's experimental batching code[1] (Li et al., 2025) encounters similar framework limitations. Supporting non-contiguous position IDs would require custom CUDA kernels for each base model (Qian et al., 2024)—a prohibitive engineering cost that sacrifices portability.

✗ **Rollback Approach (speculation waste).** After each verification step, all sequences are truncated to the batch's minimum accepted length (Wolf et al., 2020; kamilakesbi, 2024). This guarantees alignment but discards correctly verified tokens from faster sequences. As batch size grows and acceptance-rate variance widens, the waste compounds; in the extreme, one persistently rejecting sequence forces single-token progress for the entire batch. In effect, throughput collapses to that of the worst-performing sequence, undermining speculative gains and rendering the approach impractical at larger scales.

---

[1] While EAGLE's main contribution concerns improved draft models rather than batching, its repository includes experimental batch-related code we analyzed for implementation challenges. `https://github.com/SafeAILab/EAGLE/issues/250`

✓**Dynamic Padding Approach.** This approach realigns sequences after each verification by adjusting left padding to maintain right alignment, preserving all accepted tokens. While conceptually simple, correctness requires tight synchronization of position IDs, attention masks, and the KV-cache. DSD's experimental code (Yan et al., 2025) follows this idea but merely repads at each step—adding varying left padding without ever unpadding—thereby inflating sequences. It also contains three critical errors: (i) sampling bonus tokens from the draft model rather than the target model; (ii) redundantly regenerating KV-cache entries, causing memory bloat; and (iii) desynchronizing padding, position IDs, and the KV-cache across iterations. Despite the overhead of repeated realignment, a correct dynamic-padding implementation fits within standard frameworks and preserves all verified tokens.

Among the three approaches, only dynamic padding is viable: position-ID schemes require custom kernels that undermine portability, and rollback discards verified tokens at rates that grow with batch size; dynamic padding maintains correctness within standard frameworks.

**Correctness Criterion.** We formalize the correctness requirement for batch speculative decoding. Under greedy decoding (temperature $= 0$), speculative decoding must yield outputs identical to standard autoregressive generation, and any token-level divergence indicates an implementation error. For temperature $> 0$, where outputs are sampled stochastically, speculative decoding must preserve the output distribution of the target model; that is, the probability of generating any token sequence must remain unchanged. This *losslessness*, whether measured by exact output equivalence or distributional equivalence, is the defining criterion of correct speculative decoding and the hallmark of an authentic implementation. Methods that achieve high throughput while violating this criterion, as we demonstrate for BSP and DSD in Figure.1, do not constitute valid speculative decoding regardless of their reported performance metrics.                    NEW

## 3 METHOD

We present a synchronization-aware approach to batch speculative decoding that co-designs correctness and efficiency. The core tension is that preserving correctness requires synchronizing position IDs, attention masks, and the KV-cache across ragged tensors—an overhead that can consume 40% of computation. We introduce two complementary mechanisms: EQSPEC specifies and enforces the minimal synchronization invariants required for valid computation (Section 3.1), while EXSPEC groups same-length sequences to avoid synchronization overhead (Section 3.3). Together, these form a unified system in which correctness constraints drive scheduling, enabling both output equivalence and practical performance.

### 3.1 MINIMAL BATCH REALIGNMENT: EQSPEC

Figure 3 illustrates the core challenge and our remedy for maintaining correctness in batch speculative decoding. After each verification round, sequences accept different numbers of draft tokens, producing ragged tensors that GPUs cannot process. To restore a valid batch, we apply an *unpad–append–repad* procedure that converts ragged outputs back to a rectangular layout while preserving three invariants: contiguous position IDs, valid attention masks, and aligned KV-cache entries.

Correct implementation requires padding-agnostic position IDs that reset to zero at the first content token, attention masks that exclude padding tokens, and precise KV-cache realignment after each verification step (Algorithm 1). After *unpad–append–repad*, index offsets shift across sequences; consequently, position IDs and attention masks must be recomputed to preserve correct token relationships. Moreover, the bonus token introduces

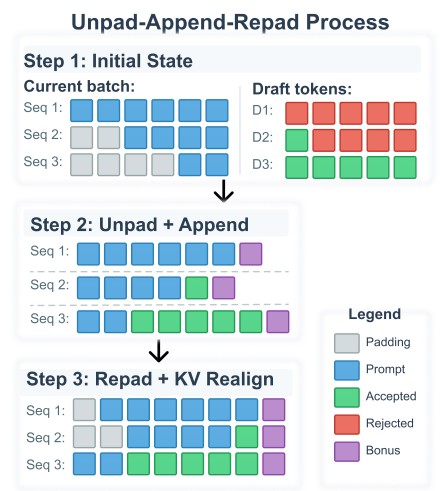

Figure 3: EQSPEC synchronizes via unpad–append–repad.

a special case: it is sampled from the target model's
output distribution at the first mismatch position after the verification forward pass has already completed. Consequently, while it encodes the target's authoritative correction, it lacks KV-cache entries in either the draft or target model because KV-cache is computed only for tokens that were present in the input sequence during the forward pass—and the bonus token, being an output, was not yet part of that input sequence. Therefore, it must be included in the next forward pass to create its KV-cache entries, further complicating synchronization. Finally, the realignment is resource-intensive: the KV-cache consists of rank-4 tensors (batch × heads × sequence × dimension), and each padding adjustment triggers allocation and concatenation of high-dimensional zeros. Algorithm 2 details the complete EQSPEC procedure.

## 3.2 THEORETICAL SPEEDUP ANALYSIS

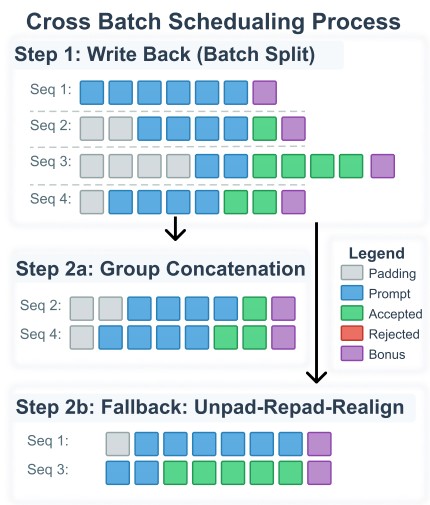

Figure 4: EXSPEC pools ragged sequences by length, avoiding realignment; only unmatched sequences need syncing, turning fixed overhead into optional cost.

FIX

The speedup of speculative decoding depends on both the token-acceptance rate and computational costs. The original formulation by Leviathan et al. (2023) analyzes single-sequence performance, modeling the expected tokens generated per iteration as $(1-\alpha^{k+1})/(1-\alpha)$, where $\alpha$ is the token acceptance rate (TAR) and $k$ is the number of draft tokens per speculation round. This single-sequence view assumes negligible batch overhead and focuses purely on acceptance dynamics.

Batch speculative decoding, however, introduces additional complexities not captured by that model—most notably alignment overhead, which can dominate and degrade performance. We therefore introduce a batch-aware speedup model:

$$ S = \frac{\alpha \cdot k}{c_{\text{draft}} + c_{\text{verify}} + c_{\text{overhead}}(B)} \quad (1) $$

where $S$ denotes speedup relative to non-speculative decoding (i.e., running the target model alone), $c_{\text{draft}}$ and $c_{\text{verify}}$ are the relative costs of draft generation and target verification, and $c_{\text{overhead}}(B)$ captures batch-dependent alignment overhead absent from single-sequence analysis.

Two observations follow. First, when sequences within a batch share the same length, batch realignment overhead is negligible; likewise, with very small draft models or prompt lookahead decoding (Fu et al., 2024b), $c_{\text{draft}}$ becomes small relative to verification. Second, and critically, $c_{\text{overhead}}(B)$ scales superlinearly with batch size $B$ due to the ragged-tensor problem (Section 4.3). This overhead decomposes as $c_{\text{overhead}}(B) = c_{\text{pad}}(B) + c_{\text{kv}}(B)$, where $c_{\text{pad}}(B)$ accounts for unpad–append–repad operations and $c_{\text{kv}}(B)$ for KV-cache realignment on rank-4 tensors. While $c_{\text{draft}}$ and $c_{\text{verify}}$ benefit from GPU parallelism and remain relatively stable within the hardware's batch-parallel regime, the alignment overhead grows with both batch size and variance in acceptance rates across sequences.

## 3.3 CROSS-BATCH SCHEDULING: EXSPEC

Profiling EQSPEC shows that alignment overhead consumes 39.4% of computation at batch size 8, rising to 46.7% at batch size 16. Because this cost is inherent to synchronizing ragged tensors, micro-optimizing the primitives yields limited gains. Instead of accelerating these operations, EXSPEC avoids them by scheduling.

EXSPEC differs from EQSPEC in how it manages sequence lifecycles. Rather than maintaining fixed batches that require realignment after every verification, we introduce a *SequencePool* that holds sequences individually in their ragged states. This enables three optimizations: (i) sequences that complete (reach EOS) are immediately removed, avoiding wasted computation on finished items; (ii) *lazy realignment* defers synchronization until strictly necessary, keeping sequences in their natural

ragged form between steps; and (iii) *dynamic batch formation* over a sliding window of $W > B$ sequences greatly increases the chance of finding same-length groups that can be concatenated without any realignment.

Figure 4 illustrates the flow. After verification, ragged sequences return directly to the pool without realignment. The scheduler then scans the active window and attempts to form batches of identical length. Same-length sequences concatenate directly—no padding adjustments, no position-ID recomputation, no KV-cache realignment—bypassing $c_{\text{overhead}}(B)$. Only when same-length grouping fails do we fall back to the expensive *unpad–append–repad* procedure from EQSPEC. Algorithm 3 formalizes this in four phases: dynamic batch formation (prioritizing same-length groups), draft generation, verification, and pool write-back with continuous window refresh. Combined with prompt-length sorting, grouping rates approach unity for similar workloads, turning the worst case (constant realignment) into the rare case.

## 4 EXPERIMENTS

We evaluate EQSPEC and EXSPEC along two dimensions often conflated in prior work: **correctness** (whether outputs match non-speculative generation) and **throughput** (tokens per second). Through systematic evaluation across three model families and comparisons with both research prototypes and production systems, we show that our approach uniquely preserves output equivalence while achieving competitive speedups.

---

**Algorithm 3** EXSPEC: Cross-Batch Scheduling

**Require:** Draft and Target model $\mathcal{M}_d, \mathcal{M}_t$, Prompts $\mathcal{P}$, Window size $W$, Batch size $B$
**Ensure:** Generated sequences $\mathcal{S}$
1: $Pool \leftarrow \texttt{InitSequencePool}(\mathcal{P})$
2:      ▷ *Tokenize and optionally sort by length*
3: $Window \leftarrow \texttt{RefillWindow}(Pool, W)$
4: **while** $Pool.hasActive()$ **do**
5:     **Phase 1: Lazy Realignment**
6:     $\mathcal{B}, mask, KV \leftarrow \texttt{GetBatch}(Window, B)$
7:          ▷ *Try same-length concatenation*
8:      ▷ *Fallback to Unpad-Repad Realignment*
9:     **Phase 2: Draft Generation**
10:    $D \leftarrow \mathcal{M}_d.\texttt{Generate}(\mathcal{B}, mask, K)$
11:    **Phase 3: Batch Verification**
12:    $A, B, KV \leftarrow \texttt{BatchVerify}(\mathcal{M}_t, \mathcal{B}, D, KV)$
13:    **Phase 4: Write-Back and Window Refill**
14:    **for** $i \in \mathcal{B}$ **do**
15:       $Pool[i] \leftarrow Pool[i] \oplus A[i] \oplus B[i]$
16:       $Pool.KV[i] \leftarrow KV[i]$
17:       **if** $isComplete(Pool[i])$ **then**
         $Pool.deactivate(i)$
18:    $Window \leftarrow \texttt{RefillWindow}(Pool, W)$
19: **return** $Pool.sequences$

---

### 4.1 EXPERIMENTAL SETUP

**Models.** To demonstrate generality, we evaluate three target–draft pairs: Vicuna-7B/68M (Zheng et al., 2023), Qwen3-8B/0.6B (Yang et al., 2025), and GLM-4-9B/0.6B (GLM et al., 2024). Unless otherwise noted, experiments use NVIDIA A100 80GB GPUs, PyTorch 2.7, HuggingFace Transformers 4.51.3, five draft tokens per speculation round, and greedy decoding for determinism.

**Evaluation and Datasets.** We use SpecBench (Xia et al., 2024), focusing on the first turn of each conversation because our evaluation targets offline batch inference rather than multi-turn interaction. We also use Multi30k (Elliott et al., 2016) for a controlled EXSPEC study that contrasts random sampling with an identical-length subset, isolating sequence-length diversity as the driver of grouping rate. For the main evaluation, we measure: (1) *Throughput*: tokens/s across batch sizes; and (2) *Correctness*: exact-match rate (full-sequence equivalence with non-speculative decoding) and partial-match rate (fraction of tokens matching until the first divergence). The partial-match metric helps localize failure modes—early divergence typically indicates position-ID or KV-cache misalignment.

**Batch Speculative Decoding Compared.** Following our design-space taxonomy (Section 2), we evaluate: (1) *Masking approaches*: **BSP** (Su et al., 2023) attempts masking with adaptive speculation but suffers position-ID inconsistencies (BASS (Qian et al., 2024) also follows this approach but requires custom CUDA kernels, limiting generality); (2) *Dynamic-padding approaches*: **DSD** (Yan et al., 2025) explores padding but mishandles the KV-cache, while **EQSPEC** implements correct synchronization and **EXSPEC** adds cross-batch scheduling; (3) *Reference baselines*: **Spec-1** (batch-size-1 speculation from Hugging Face Transformers), which does not support batch speculative

| Method | Vicuna | | | | Qwen3 | | | | GLM4 | | | |
|---|---|---|---|---|---|---|---|---|---|---|---|---|
| | Batch 1 | | Batch 4 | | Batch 1 | | Batch 4 | | Batch 1 | | Batch 4 | |
| | E | P | E | P | E | P | E | P | E | P | E | P |
| *Batch-based Methods (vs. batch=1 non-spec)* | | | | | | | | | | | | |
| Non-Spec-Batch | – | – | 53.8 | 98.2 | – | – | 92.9 | 96.5 | – | – | 93.3 | 97.2 |
| Spec-1 | 97.1 | 98.4 | – | – | 94.6 | 97.2 | – | – | 96.0 | 98.0 | – | – |
| EQSPEC | 97.3 | 98.6 | 92.1 | 98.6 | 94.6 | 96.9 | 92.3 | 95.7 | 96.7 | 98.1 | 96.5 | 98.3 |
| EXSPEC | 97.3 | 98.6 | 90.8 | 97.6 | 94.6 | 96.9 | 95.0 | 97.1 | 96.7 | 98.1 | 95.2 | 97.7 |
| DSD | 0.0 | 8.1 | 0.0 | 2.2 | 0.2 | 2.2 | 0.0 | 0.6 | 0.0 | 1.0 | 0.0 | 0.8 |
| BSP | 1.9 | 39.7 | 0.2 | 31.3 | 3.5 | 19.9 | 2.1 | 12.6 | 1.0 | 15.3 | 0.6 | 8.1 |
| *Continuous Batching Systems (vs. own non-spec)* | | | | | | | | | | | | |
| vLLM + Spec | 96.9 / 98.0 | | | | 65.6 / 78.5 | | | | 72.7 / 84.5 | | | |
| SGLang + EAGLE | 69.8 / 79.5 | | | | 47.7 / 65.4 | | | | – | | | |
| SGLang + EAGLE + Det | 85.0 / 90.0 | | | | 50.6 / 69.5 | | | | – | | | |

Table 1: Correctness check reports exact match (E) and partial match (P) scores. Top: compares with batch=1 non-spec baseline. Bottom: with each system's non-speculative mode (scores reported as E / P). Our approach sustains $> 95\%$ accuracy, while prior work (DSD, BSP) suffers major drops from KV-cache and position ID errors.

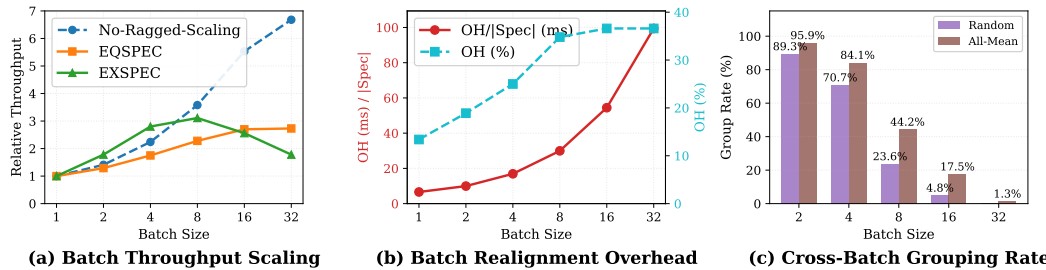

| (a) Batch Throughput Scaling | (b) Batch Realignment Overhead | (c) Cross-Batch Grouping Rate |
|---|---|---|

Figure 5: Decomposing batch speculative decoding performance. (a) Batch scaling efficiency: each method's throughput normalized to its own BS=1 baseline, isolating scaling behavior from absolute performance. The No-Ragged-Scaling line shows measured autoregressive decoding throughput (not a theoretical bound), serving as a reference for how standard batching scales without ragged tensor overhead. (b) Alignment overhead grows super-linearly with batch size, consuming up to 38% of inference time, validating that $c_{\text{overhead}}(B)$ dominates at scale. (c) Cross-batch grouping rates on Multi30k for random vs. uniform-length sequences, showing that length homogeneity transforms grouping effectiveness.

decoding[2]. We also compare with production systems **vLLM**[3] (Kwon et al., 2023) (which subsumes TETRIS (Wu et al., 2025) and TurboSpec (Liu et al., 2025) as vLLM forks) and **SGLang-EAGLE**[4] (Zheng et al., 2024; Li et al., 2024b; 2025), noting that their continuous batching and memory-management designs complicate direct comparison. We further test **SGLang-EAGLE-Deterministic** (SGLang Team, 2025), which enables deterministic execution to reduce numerical variance. No existing implementation uses rollback due to its inherent wastefulness.

## 4.2 OUTPUT CORRECTNESS VERIFICATION

We verify correctness using deterministic greedy decoding to eliminate sampling variance and enable precise bug isolation. This avoids metrics such as ROUGE (Qian et al., 2024), which can mask implementation failures (e.g., repetitive corruption can still score reasonably). Instead, we use exact match (any divergence) and partial match (fraction of tokens before the first mismatch) to diagnose failure modes.

---

[2]https://github.com/huggingface/transformers/issues/32165

[3]We use the vLLM v0 engine because v1 deprecates speculative decoding.

[4]SGLang is compatible only with the EAGLE family; we compare Vicuna-7B/EAGLE2 and Qwen3-8B/EAGLE3. There are no available weights for GLM-4.

Table 1 reveals distinct patterns. Our methods maintain ≈95% exact match across settings, the remaining ≈5% divergence stems from numerical non-determinism in floating-point operations (He & Lab, 2025) and tie-breaking in argmax sampling rather than algorithmic errors. By contrast DSD and BSP fail catastrophically with different signatures. DSD's near-zero scores indicate immediate position-ID misalignment—the model fails from the first token. BSP shows higher partial match (up to 39.7%) but low exact match, indicating gradual degradation: outputs are initially correct before KV-cache drift misdirects attention, triggering repetition. These complementary patterns—immediate failure vs. gradual decay—show how partial-match metrics reveal not just *that* implementations fail, but *when* and *why*.

Production systems (vLLM, SGLang) introduce additional complexity. While both are accurate on Vicuna, they degrade markedly on Qwen3. SGLang-EAGLE-Deterministic helps disambiguate the cause: improved accuracy suggests most divergences stem from floating-point non-determinism (He & Lab, 2025) rather than algorithmic bugs. Despite this, we show below that their throughput still falls below non-speculative baselines.

### 4.3 OVERHEAD AND SCALING DYNAMICS

We now validate our theoretical predictions through five studies that decompose batch speculative decoding performance. These experiments isolate alignment overhead growth, quantify its impact on batch scaling, identify sequence diversity as the key bottleneck to grouping effectiveness, compare against production systems, and provide mechanistic insights through overhead profiling.

**Batch scaling efficiency.** Figure 1 shows that EXSPEC outperforms EQSPEC in absolute throughput by combining speculative and batching gains, yet EQSPEC exhibits negative scaling beyond BS=8. To test whether batch speculative decoding can retain GPU-parallelism benefits despite raggedness, Figure 5(a) measures *batch scaling efficiency*: each method's throughput at batch size $N$ divided by its own throughput at batch size 1. This normalization isolates how well each method scales with batch size, independent of absolute performance differences. The *No-Ragged-Scaling* line represents actual measured autoregressive decoding throughput (without speculation), similarly normalized to its own BS=1 baseline; it is not a theoretical upper bound but rather a reference showing how standard batching scales when no ragged tensor synchronization is required. A notable effect emerges: EXSPEC initially exceeds this reference line. This occurs because each method is normalized independently, and speculation converts memory-bound token generation into compute-bound verification, which benefits more from GPU parallelism. Although EXSPEC may have lower absolute throughput than standard decoding at BS=1 due to draft model overhead, its relative scaling from BS=1 to BS=8 can surpass that of non-speculative decoding. However, at larger batch sizes, alignment overhead dominates and the advantage inverts, confirming that $c_{\text{overhead}}(B)$ eventually overwhelms parallelism gains. The key finding is that correct batch speculative decoding need not sacrifice batch scaling efficiency, given that prior methods either produce incorrect outputs or fail to scale.

FIX

**Alignment overhead growth.** Figure 5(b) quantifies realignment costs via two metrics: percentage of total time spent on alignment (OH%) and per-round alignment time (OH/|Spec|). Overhead rises from ∼ 13% at BS=1 to nearly 40% at BS=32, with per-round costs increasing even more. This matches our prediction that $c_{\text{pad}}(B) + c_{\text{kv}}(B)$ grows super-linearly with $B$. Crucially, this is not merely an implementation inefficiency: the very operations required for correctness (unpad–append–repad and KV-cache realignment) become increasingly expensive as sequence lengths diverge.

**Grouping rate × sequence-length distribution.** Figure 5(c) probes whether cross-batch scheduling limits are algorithmic or circumstantial via Multi30k. Comparing random sampling to an All-Mean subset with identical lengths isolates the bottleneck: sequence diversity, not the method. Random sampling shows grouping rates collapsing with batch size, whereas the All-Mean configuration maintains high grouping effectiveness even at moderate scales, substantially reducing $c_{\text{overhead}}(B)$. This contrast suggests that preprocessing strategies (e.g., bucketing, dynamic sorting) can push real workloads toward the ideal, revealing untapped potential when scheduling is paired with workload shaping.

| Method | TPS | \|Spec\| | Time/Verif. (ms) | Verif. % | Draft % | Overhead % |
|--------|-----|----------|------------------|----------|---------|------------|
| EQSPEC | 95.6 | 1469 | 24.88 | 44.9 | 27.4 | 27.7 |
| EXSPEC | 156.4 | 952 | 29.13 | 55.9 | 29.5 | 14.6 |

Table 2: Overhead anatomy of batch speculation methods. Despite slower per-verification time, EXSPEC achieves higher throughput by reducing total verification calls and minimizing alignment overhead through intelligent scheduling.

| Batch Size | Method | Throughput | P50 | P90 | P99 |
|------------|--------|------------|-----|-----|-----|
| 1 | EQSPEC | 15.20 | 6.70 | 12.96 | 16.53 |
| 1 | EXSPEC | 15.78 | 6.08 | 8.32 | 9.42 |
| 2 | EQSPEC | 26.70 | 7.96 | 14.78 | 19.06 |
| 2 | EXSPEC | 30.54 | 9.75 | **114.89** | **134.03** |
| 4 | EQSPEC | 46.11 | 9.46 | 16.23 | 19.13 |
| 4 | EXSPEC | 52.44 | 8.01 | **53.21** | **70.35** |
| 8 | EQSPEC | 76.03 | 11.26 | 19.23 | 22.50 |
| 8 | EXSPEC | 77.54 | 9.16 | 19.13 | **33.93** |

Table 3: Latency-throughput tradeoff under simulated online serving with heterogeneous request lengths. Throughput is in tokens/s. P50, P90, and P99 denote the 50th, 90th, and 99th percentiles of request completion latency, respectively. Bold values indicate significantly worse tail latencies for EXSPEC.

**Overhead Anatomy.** Table 2 shows that EXSPEC attains higher throughput via a deliberate trade-off. Cross-batch scheduling cuts total verification calls by one-third and halves alignment overhead by grouping same-length sequences. The trade-off is memory locality: dynamic batching scatters KV-cache entries, increasing per-verification latency. Despite slower individual operations, the reduction in operation count yields a 64% overall throughput gain. This balance between operation count and efficiency suggests future work on improving KV-cache locality within the cross-batch framework.

## 4.4 LATENCY-THROUGHPUT TRADEOFF IN ONLINE SERVING

To evaluate performance under realistic online serving conditions, we conducted experiments simulating dynamic request arrivals using full multi-turn conversations from SpecBench. We shuffled conversation turns to maximize length diversity and inserted new turns into the next available batch, measuring both throughput (tokens/s) and request completion latency (s) at P50/P90/P99 percentiles. Table 3 compares EQSPEC against EXSPEC across batch sizes 1, 2, 4, and 8.                NEW

Both methods achieve positive speedups under heterogeneous multi-turn workloads, with EXSPEC obtaining 2–14% higher throughput than EQSPEC through cross-batch grouping of same-length sequences. However, EXSPEC suffers 1.5–7.7× worse P90/P99 latencies when requests are delayed to enable grouping, as early requests must wait for later ones with matching lengths. This latency penalty is particularly severe at smaller batch sizes where grouping success rates are low (23.6% at batch size 2, as shown in Figure 5 c), causing head-of-line blocking that inflates tail latencies to over 100 seconds. In contrast, EQSPEC maintains predictable tail latency with P99 under 23 seconds across all batch sizes, making it suitable for latency-sensitive online serving where service-level objectives must be met. These results demonstrate that the choice between EQSPEC and EXSPEC depends on deployment requirements: EQSPEC provides stable, predictable latency for interactive applications, while EXSPEC maximizes throughput for offline batch processing where individual request latency is less critical. By offering both algorithms with explicit correctness guarantees, our work enables practitioners to match their batch speculation strategy to their specific operational constraints.                NEW

## 5 RELATED WORK

**Speculative Decoding.**    Speculative decoding accelerates LLM inference by verifying draft tokens in parallel. Two verification paradigms dominate: *sequence verification*—as in SpecDec and follow-ups (Xia et al., 2023; Santilli et al., 2023; Yang et al., 2023; Hooper et al., 2023; Zhang et al., 2023; Fu et al., 2024a)—which preserves the target model's output distribution; and *tree verification*—e.g., Medusa/EAGLE variants (Miao et al., 2024; Spector & Re, 2023; Sun et al., 2023; He et al., 2023; Cai et al., 2024; Li et al., 2024a;b; 2025)—which explores multiple branches to raise acceptance rates. Recent works parallelize multiple draft sequences per request (Stewart et al., 2024; Lee et al., 2024) but still verify each request independently. Approximate schemes (Kim et al., 2023; Zhong et al., 2025) trade exactness for speed; our work assumes lossless verification to detect implementation errors rather than approximation artifacts.

**Batch Speculative Decoding.**    Extending speculative decoding to batch settings introduces the ragged tensor problem: when sequences accept different numbers of draft tokens, the resulting variable-length tensors break GPU-friendly rectangular operations (Qian et al., 2024). Existing approaches face fundamental limitations detailed in Section 2. Position ID/masking approaches like BSP (Su et al., 2023) suffer from position ID inconsistencies across iterations. Dynamic padding approaches reveal critical implementation errors: both DSD (Yan et al., 2025) and Meta's recent work (Tang et al., 2025) incorrectly sample bonus tokens from the draft model's distribution rather than the target model's, violating the fundamental correctness guarantee of speculative decoding. This error compounds with improper KV-cache handling, producing corrupted outputs—BSP generates repetitive tokens while DSD produces <unk> symbols. BASS (Qian et al., 2024) sidesteps these issues through custom CUDA kernels but sacrifices portability. These failures demonstrate that the ragged tensor problem extends beyond performance to correctness itself, motivating our cross-batch scheduling approach that maintains correctness while achieving batch scaling without custom kernels.

**Production Systems for Speculative Decoding.**    Serving stacks integrate speculation atop general batching (vLLM, SGLang/EAGLE, and forks such as TETRIS and TurboSpec (Kwon et al., 2023; Zheng et al., 2024; Li et al., 2025; Wu et al., 2025; Liu et al., 2025)). In practice, variable draft acceptance clashes with continuous batching and paged attention, and our measurements indicate scaling limits (e.g., small single-sequence gains that reverse at higher concurrency); some engines have since reduced or deprecated speculative-decoding support. We intentionally build our reference implementation from scratch, prioritizing correctness checks; compatibility with continuous batching, paged attention, and prefill–decode separation is orthogonal and left as future work.

## 6 CONCLUSION

We presented a correctness-first approach to batch speculative decoding that resolves the fundamental tension between maintaining output equivalence and achieving practical speedups. By identifying the precise synchronization invariants required for correctness, we explained why existing methods fail and established the minimal requirements for valid computation. In particular, our EQSPEC implementation enforces these invariants but reveals that realignment overhead consumes up to 40% of computation. By contrast, EXSPEC overcomes this limitation via cross-batch scheduling that dynamically groups same-length sequences to bypass realignment entirely. Moreover, experiments across three model families show that our approach uniquely preserves >95% output equivalence while achieving up to 3× throughput improvement at batch size 8, hereby validating that a careful co-design of correctness constraints and scheduling can multiply batch parallelism with per-sequence speculation gains. Our findings on ragged shape tensor synchronization overhead and cross-batch scheduling may inform future designs for production systems, where integrating speculation with continuous batching remains an open challenge.

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

## A    APPENDIX

### BATCH SCALING OF SPECULATIVE DECODING IN VLLM AND SGLANG

FIX

**Production systems limitation.**    Figure 6 shows that production frameworks struggle with batch speculative decoding. Although our method is not directly comparable to continuous-batching systems (which vary effective batch size dynamically), the trend is consistent: vLLM's speculative decoding underperforms its baseline at high concurrency, and SGLang+EAGLE is slower than non-speculative generation across all batch sizes. vLLM's v0 engine previously used batch expansion to sidestep the ragged tensor problem: instead of verifying draft tokens [1, 2, 3] together and handling variable acceptance lengths, it duplicated each sequence K times with variants [1], [1,2], [1,2,3], verified all variants in one pass, then kept only the longest correct prefix. This approach was deprecated in v1 (vllm-project/vllm#17984) because it wastes $K\times$ memory and $K\times$ compute—if only token 1 is accepted, the longer variants consumed resources for nothing. This caused GPU memory overflow at scale and broke CUDA graph compatibility.

FIX

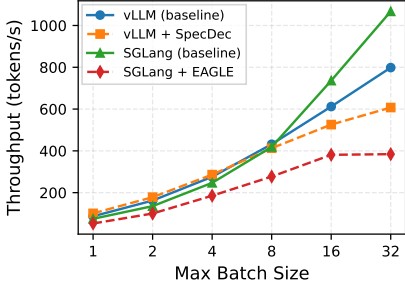

Figure 6: Speculative decoding lags non-speculative baselines, with larger batches further degrading throughput.

### DISCUSSION

NEW

Batch parallelism and per-sequence speculative decoding represent two orthogonal acceleration strategies that should, in principle, multiply together: batching exploits GPU parallelism across sequences while speculation reduces sequential token dependencies within each sequence. However, when combining these techniques in practice, the ragged tensor problem emerges as a fundamental obstacle that breaks this multiplicative relationship. Different sequences accept different numbers of draft tokens, desynchronizing position IDs, attention masks, and KV-cache state across the batch. A correctly implemented batch speculative decoding, as we provide through EQSPEC and EXSPEC, preserves the multiplicative gains but incurs an inherent realignment cost that consumes up to 40% of computation. Critically, batch speculation amplifies per-sequence gains; it cannot create them.

If a draft-target pair shows no speedup at batch size 1 due to low acceptance rates or high draft overhead, batching cannot recover what does not exist at the single-sequence level. Our work establishes the synchronization invariants required for correctness, quantifies their irreducible costs, and demonstrates through cross-batch scheduling that intelligent system design can mitigate, though not eliminate, the overhead of maintaining output equivalence at scale.                    NEW

## REPRODUCIBILITY STATEMENT

We provide our complete implementation in the Supplementary Material, including all hyperparameters, experimental configurations, and model specifications detailed in Section 4. Our correctness verification framework using exact and partial match metrics enables deterministic validation of both our results and future implementations. All experiments use publicly available models and datasets for reproducibility.

**Model Configurations.**    We evaluate three target-draft model pairs: Vicuna-7B (lmsys/vicuna-7b-v1.3) with a 68M draft model (double7/vicuna-68m), Qwen3-8B (Qwen/Qwen3-8B) with Qwen3-0.6B (Qwen/Qwen3-0.6B), and GLM-4-9B (zai-org/GLM-4-9B-0414) with a 0.6B draft model (jukofyork/GLM-4.5-DRAFT-0.6B-v3.0). All models were loaded in FP16 precision with greedy decoding (temperature=0, top_p=1.0) to ensure deterministic outputs. We use five draft tokens per speculation round across all experiments unless otherwise specified.

**Production Systems Configuration.**    We evaluated two production inference systems for comparison: vLLM version 0.9.1 and SGLang commit c4e314f (the deterministic decoding had not merged to the stable version at the time of submission, so we compiled it from source). For vLLM, we used the V0 engine with speculative decoding enabled, as the V1 engine does not support draft model speculative decoding[5]. For SGLang, we used EAGLE-based speculation with model-specific draft models: yuhuili/EAGLE-Vicuna-7B-v1.3 for Vicuna-7B and Tengyunw/qwen3_8b_eagle3 for Qwen3-8B. GLM-4 was not evaluated with SGLang due to the unavailability of compatible EAGLE draft models. We tested both standard SGLang inference and SGLang with deterministic mode enabled (SGLang Team, 2025) to isolate floating-point non-determinism from algorithmic correctness issues.

## DISCLOSE ON LLM USAGE

We used Large Language Models as assistive tools in preparing this manuscript: GPT-5 and Claude Opus 4.1 for polishing writing (grammar correction and sentence restructuring), generating conceptual diagram illustrations, and writing unit tests; NVIDIA/Parakeet-TDT-0.6B-v3 for voice input transcription. LLMs were not used for research ideation, experimental design, data analysis, or scientific conclusions. All core algorithmic implementations and scientific contributions are solely from the authors. We take full responsibility for all content in this paper.

## ETHICS STATEMENT

This work focuses on improving the efficiency of LLM inference without altering model outputs or introducing biases. Our correctness-preserving approach ensures that acceleration techniques do not compromise model safety or reliability. Our open-source release enables reproducible research and transparent evaluation of batch speculative decoding techniques.

---

[5]https://github.com/vllm-project/vllm/issues/21797

