# OpenReview forum: "Batch Speculative Decoding Done Right"
_ICLR.cc/2026/Conference — Submitted to ICLR 2026_

### Official Review · Reviewer_RaXj · 2025-10-30

**Soundness:** 2
**Presentation:** 3
**Contribution:** 3
**Rating:** 4
**Confidence:** 4

**Summary:**

This paper proposes EQSpec and EXSpec, focusing on the correctness of batch speculative decoding. The paper claims that existing speculative decoding systems fail to preserve output correctness (aligned to the original base model) when batch size > 1, due to corrupted position IDs, attention masks, and KV cache state. To solve this, it proposes a group-and-padding batch scheduling algorithm to correctly synchronizing among the ragged tensors in a batch. Experimental results show that EXSpec can largely preserve the correctness (over 90%) while other systems can merely do so.

**Strengths:**

The topic is highly related to practical LLM usage. Batch speculative decoding is an important direction, yet there has been few existing works focusing on the correctness, mostly on speed performances.

The group-then-padding algorithm is practically useful. It can mitigate the length misalignment in an efficient way. Specifically, as stated in Section2, it does not involve modifying position IDs, avoiding re-implementing a whole new kernel, and also preserve the accepted tokens from being cropped.

The demonstration figures (Fig.3 and 4) are quite informative, making the main design easily understood.

**Weaknesses:**

Main concerns:

As the paper stated, the problem of current inference systems is about incorrect output, which is caused by, alleged, KV-cache and position-ID errors. I think the root causes should be more specified and quantified. Is it because current systems have not implemented batch SD supports, or the implementation is incorrect, or just float precision is not accurate enough? Specifically, vLLM can achieve high match accuracy on Vicuna, but lower on other models. If the cause is about missing or incorrect implementation, I think the results would be uniformly low. If the authors provide detailed implementation or code examples, this concern will be clarified.

The paper claims that the output will be corrupted at batch size > 1, but Table 1 shows that BSP and DSD also have significantly low accuracy when batch size=1. That is a misalignment between the claim and evidence. Furthermore, this result also indicates that the problem is not about batch size, but other factors, while the optimization method largely targets at batch size > 1.

Minor concerns:

The baseline of token throughput in Fig.5(a) is ‘no speculation’. ExSpec only achieves marginal acceleration compared to the original auto-regressive decoding, which is slow. As a speculative method, it should be compared to other speculative-decoding baselines for token throughput.

The claim that ‘speculative decoding needs to yield identical output’ can be more accurate: it is only true when temperature=0, while for temperature>0 the output of base model is a distribution and the output tokens are sampled from this distribution, so there is basically no ‘correct output’ but only a distribution.

The introduction and experiment analysis are hard to read. Improvement on writing would be beneficial.

**Questions:**

1. Could you provide details about the cause of corrupted output of existing systems , e.g. missing or incorrect implementation, to further clarify the cause of corrupted outputs?
2. Does the incorrectness also exist when batch size = 1? If so, how does the proposed method address this issue, given that the modification is only about batch scheduling?
3. How does the methods perform compared to speculation-based baselines in terms of inference speed?

---

> ### Author Response · Authors · 2025-11-21
>
> We thank the reviewer for the thoughtful and constructive feedback!
>
> ## Weakness 1 & Question 1 and 2
> > As the paper stated, the problem of current inference systems is about incorrect output, which is caused by, alleged, KV-cache and position-ID errors. I think the root causes should be more specified and quantified. Is it because current systems have not implemented batch SD supports, or the implementation is incorrect, or just float precision is not accurate enough? Specifically, vLLM can achieve high match accuracy on Vicuna, but lower on other models. If the cause is about missing or incorrect implementation, I think the results would be uniformly low. If the authors provide detailed implementation or code examples, this concern will be clarified.
>
> > The paper claims that the output will be corrupted at batch size > 1, but Table 1 shows that BSP and DSD also have significantly low accuracy when batch size=1. That is a misalignment between the claim and evidence. Furthermore, this result also indicates that the problem is not about batch size, but other factors, while the optimization method largely targets at batch size > 1.
>
> > Could you provide details about the cause of corrupted output of existing systems to further clarify the cause of corrupted outputs? Does the incorrectness also exist when batch size = 1?
>
> **Response:**
>
> **Short Answer**: We thank the reviewer for this careful observation. You've identified a key distinction we need to clarify. **BSP and DSD show correctness failures at both BS=1 and BS>1 in Table 1**, stemming from multiple implementation issues, not just ragged tensor handling. **We’ve fixed both types of the issues.** For vLLM and SGLang, the situation is different. vLLM's batch expansion approach is fundamentally wasteful (we’ll elaborate this in the detailed explanation), but we cannot definitively diagnose their specific correctness issues—we only observe low accuracy  and negative speedups even with deterministic execution patch [1] (SGLang+EAGLE+Det: 50.6% on Qwen3). Our contribution establishes the minimal synchronization requirements for correctness at any batch size, providing a principled foundation that clarifies what correct batch speculative decoding requires.
>
> **Detailed Explanation**: You are correct that Table 1 shows BSP and DSD failing at both BS=1 and BS>1. Let us clarify: **the ragged tensor problem is specifically about BS>1, but BSP and DSD have multiple implementation bugs, some of which break correctness at any batch size.**These implementations suffer from **two categories of bugs**, and **we’ve fixed both types of issues**:
>
> | Bug Category | Specific Issue | System | Affects BS=1? | Affects BS>1? |
> |--------------|----------------|--------|---------------|---------------|
> | Batch-independent implementation errors  | Bonus token sampled from draft model instead of target model | DSD | ✓ | ✓ |
> | | KV cache contains rejected tokens; position IDs auto-calculated incorrectly | BSP | ✓ | ✓ |
> | Ragged tensor synchronization (batch-specific) | Unaligned KV cache across sequences with different accepted lengths | DSD, BSP | ✗ | ✓ |
> | | Position ID desynchronization across ragged sequences | BSP | ✗ | ✓ |
>
> **The first category** consists of relatively straightforward implementation mistakes—sampling from the wrong distribution or failing to exclude rejected tokens. While these corrupt outputs at any batch size, they could be fixed with careful code review.
>
> DSD: Bonus token from draft model:
>
> ```python
> matched_tokens = first_false_positions + torch.ones_like(first_false_positions)
>
> # WRONG: All matched_tokens from draft model output
> input_list.append(torch.cat((padding, input_tensors[idx],
>                              cat_tensor[idx][:matched_tokens[idx]])))
>
> # CORRECT: N draft tokens + 1 oracle bonus token from the target model
> input_list.append(torch.cat((padding, input_tensors[idx],
>                              cat_tensor[idx][:first_false_positions[idx]],
>                              bonus_tokens[idx].unsqueeze(0))))
> ```
> BSP: KV cache with rejected tokens
> ```python
> while True:
>     # Verify speculated tokens with target model
>     verify_inputs = torch.cat([first_token, speculated_tokens], axis=1)  # K+1 tokens
>     logits, ret.past_key_values =  self.model(verify_inputs, attention_mask=attention_mask,  use_cache=True, past_key_values=kv_cache)
>     # ✗ BUG: Contains KV for ALL K+1 tokens
>
>     # Find which tokens are correct
>     check_mask = torch.cumsum(correct == speculated_tokens, 1) == torch.arange(1, specualtive_step + 1)
>     correct_len = torch.sum(check_mask, axis=1)  # e.g., only 2 out of 5 matched
>
>     # Mask out rejected tokens in attention
>     attention_mask[:, -specualtive_step:] = check_mask  # Zeros out rejected positions
>
> # Next iteration:
> # ✗ CRITICAL MISMATCH: kv_cache has length for 5 tokens, but attention_mask only has 2!
> # ✗ Position IDs auto-calculated from attention_mask will be wrong!
> ```
> (see next)

---

> > ### Author Response · Authors · 2025-11-21
> >
> > ## Weakness 1 & Question 1 and 2 (continued)
> >
> > **The second category, ragged tensor synchronization, is the fundamental, systematically overlooked challenge that our paper addresses.**
> >
> > Wrong: Both BSP and DSD:
> >
> > **Category 2: Ragged Tensor Synchronization (The Hard Problem)**
> > ```python
> > # missing re-aligned cache_position and position_ids.
> > outputs = target_model(verify_inputs, attention_mask=attention_mask, use_cache=True, past_key_values=kv_cache)
> >
> > ```
> > Our Correct Implementation (EQSPEC):
> > ```python
> > # cache_position indicates the starting and ending position of draft_tokens_tensor for paralleled verification.
> > cache_position = torch.arange(input_tensors.shape[1] - 1, input_tensors.shape[1] + draft_tokens_tensors.shape[1], device=device)
> > # position_ids is about the correct application of the RoPE embedding.
> > start_pos = input_tensors.shape[1] - 1
> > end_pos = input_tensors.shape[1] + draft_tokens_tensors.shape[1]
> > position_ids = full_position_ids[:, start_pos:end_pos]
> >
> > # Realign AND trim the KV cache
> > target_past_key_values = realign_kv_cache(target_model, target_past_key_values, original_content_lengths, new_padding_lengths, old_padding_lengths, matched_tokens)
> >
> > outputs = target_model(torch.cat([input_tensors[:, -1:], draft_tokens_tensors], dim=1), attention_mask=attention_mask, past_key_values=realigned_kv_cache, cache_position=cache_position, position_ids=position_ids)
> > ```
> >
> > **vLLM's ragged shape problem:** vLLM's v0 engine bypass the ragged tensor problem through batch expansion: for draft tokens [1, 2, 3], it creates K=3 duplicate sequences with variants [1], [1,2], [1,2,3], verifies all in one pass, then keeps only the longest correct prefix. If only token 1 is correct, the [1,2] and [1,2,3] variants are discarded—but their KV-cache entries already consumed K× memory and K× attention computation. This brute-force approach trades the ragged tensor problem for resource explosion, causing GPU memory overflow at scale and CUDA graph incompatibility, leading to deprecation ([vllm-project/vllm#17984](https://github.com/vllm-project/vllm/issues/17984)). Our approach is fundamentally different: we identify that realignment is necessary for correctness and reduce its frequency through intelligent scheduling rather than expanding memory to avoid it.
> >
> > **Why vLLM/SGLang accuracy varies by model:** vLLM/SGLang's accuracy varies mysteriously by model. To fairly evaluate inference engine frameworks, we applied state-of-the-art deconfounding: Table 1's SGLang+EAGLE+Det uses deterministic execution [1] to eliminate floating-point non-determinism. Yet even with this patch, Qwen3 achieves only 50.6% exact match—indicating **remaining correctness issues beyond floating-point non-determinism**. Diagnosing such issues in production systems requires substantial resources: Anthropic's postmortem documents months of engineering effort to isolate subtle batching-related bugs [2]. **Our correctness-first approach offers an alternative**: establish explicit synchronization invariants that guarantee correctness by construction, rather than debugging emergent interactions in opaque systems. We note that our current implementation uses the HuggingFace Transformers framework and does not integrate with vLLM or SGLang; incorporating our algorithm into production continuous-batching systems remains important future work. Our compatibility with standard Transformers interfaces ensures our approach can serve as a foundation for such integration efforts.
> >
> > **Takeaway:** The root causes remain unclear in vLLM and SGLang. Moreover, vLLM's batch expansion approach (as discussed above) attempts to bypass the ragged tensor problem through K× resource duplication—a fundamentally wasteful strategy that led to its deprecation. Our contribution is different: we address the ragged tensor problem through explicit synchronization invariants, establishing correctness by construction using standard HuggingFace Transformers. This provides a verifiable foundation that future work can integrate into production systems.
> >
> > [1] Horace He and Thinking Machine. Defeating Nondeterminism in LLM Inference. Technical report 2025. https://thinkingmachines.ai/blog/defeating-nondeterminism-in-llm-inference/
> >
> > [2] Anthropics.  A postmortem of three recent issues. Technical report 2025. https://www.anthropic.com/engineering/a-postmortem-of-three-recent-issues

---

> > > ### Author Response · Authors · 2025-11-21
> > >
> > > ## Weakness 2 & Question 3
> > > > The baseline of token throughput in Fig.5(a) is ‘no speculation’. ExSpec only achieves marginal acceleration compared to the original auto-regressive decoding, which is slow. As a speculative method, it should be compared to other speculative-decoding baselines for token throughput.
> > >
> > > > How does the methods perform compared to speculation-based baselines in terms of inference speed?
> > >
> > > **Response**:
> > >
> > > We believe there is a misunderstanding about Figure 5(a). This figure normalizes throughput to each method's batch-size-1 baseline to analyze batch scaling efficiency, not absolute performance. **Critically, the "No-Ragged-Scaling" line is a theoretical upper bound for batch scaling, not a baseline we compare against.** The "No-Ragged-Scaling" line is the ​​standard batched inference without speculation. This represents a theoretical upper bound for batch scaling because all sequences remain aligned and decode just one token, with no synchronization overhead.  EXSPEC sometimes exceeds this bound because speculation converts memory-bound operations into compute-bound verification, improving GPU utilization.
> > >
> > > Regarding comparison to "other speculative-decoding baselines": **we have compared against all available batch speculative decoding implementations**. In Figure 1, we have compared against BSP, DSD, and Spec-1 (HuggingFace Transformers speculative decoding, which only supports batch size 1). For vLLM+Spec and SGLang+EAGLE, although we do not directly compare with them because they use continuous batching, we have conducted thorough experiments demonstrating their scaling issues (Figure 6). **There exist no other correct batch speculation baselines to compare against.** Establishing a correct batch speculative decoding algorithm is precisely our contribution.
> > >
> > > ## Weakness 3
> > > > The claim that ‘speculative decoding needs to yield identical output’ can be more accurate: it is only true when temperature=0, while for temperature>0 the output of base model is a distribution and the output tokens are sampled from this distribution, so there is basically no ‘correct output’ but only a distribution.
> > >
> > > **Response**:
> > >
> > > We acknowledge this clarification, and we will update our claim in our updated version. Our correctness claim holds rigorously under greedy decoding (temperature=0), which is standard practice for deterministic validation in speculative decoding research. For temperature>0, the correctness requirement extends naturally: speculative decoding must preserve the **output distribution** of the target model. Our verification procedure maintains this guarantee—the target model's logits determine acceptance/rejection and bonus token sampling at every step, ensuring distribution equivalence.
> > >
> > > ## Weakness 4
> > > > The introduction and experiment analysis are hard to read. Improvement on writing would be beneficial.
> > >
> > > **Response**:
> > >
> > > We thank the reviewer for this feedback and will improve clarity in the updated version, particularly in the introduction and experimental analysis sections.

---

> ### Comment · Reviewer_RaXj · 2025-11-24
>
> Thank you for preparing the rebuttal. I have some following questions:
>
> 1. As you stated, BSP and DSD have implementation failures substantially, not related to bsz > 1, and you cannot identify the problem of vLLM. This makes the contribution somehow vague: the method is about correctness of batch speculative decoding, while the problem of existing methods is either not just about batch size (BSP and DSD), or cannot be identified (vLLM).
>
> 2. Is the current method of vLLM wasteful in terms of memory, correctness or speed? Can it eliminate the ragged-tensor problem and therefore trade the batch-alignment time with memory cost?
>
> 3. The explaination of the "No-Ragged-Scaling" line even confuses me. My current understanding is that, the datapoint at bsz=1 is the speed of EQSpec and EXSpec at bsz=1 (they are equal because bsz is 1), and when bsz>1 the data of this line just increases exponentially, and the data is not a tested auto-regressive batch decoding speed. Am I correct?
>
> If so, please (1) provide the acceleration ratio of your method compared to auto-regressive decoding under different batches, which I did not find in the paper, and (2) explain why EXSpec can be even faster than the line, while I think as bsz grows the acceleration ratio must drop below the line. ( The line has already been an upper bound of **speculation** speed, not **auto-regressive** speed, so I think under the same configuration the speed cannot break the upper bound, as the redundant GPU computation resources will be further consumed, leaving less for speculative decoding to use. )

---

> > ### Author Response · Authors · 2025-11-27
> >
> > > Q1: As you stated, BSP and DSD have implementation failures substantially, not related to bsz > 1, and you cannot identify the problem of vLLM. This makes the contribution somehow vague: the method is about correctness of batch speculative decoding, while the problem of existing methods is either not just about batch size (BSP and DSD), or cannot be identified (vLLM).
> >
> > Response to Q1 (clarifying scope, fixing the misunderstanding, and contributions)
> >
> > ## What we meant vs. what was heard
> >
> > Our aim is not merely to “improve correctness a bit.” Our claim is that prior Batch Speculative Decoding implementations are **not** authentic Batch Speculative Decoding, and that  our paper is so far as we know **the first and only** method that achieves **lossless batch speculative decoding**, satisfying the true synchronization invariants. We also analyze the inherent cost of doing Batch Speculative Decoding and show how to reduce that cost without breaking equivalence. We will update our claim in our updated version.
> >
> > ## Our two main contributions
> >
> > **(C1) The first and only lossless Batch Speculative Decoding whose output distribution remain identical to the non-spec target model.**
> >
> > To our knowledge, EQSPEC is the **first and only** method of lossless batch speculative decoding, whose batched output distribution remain **identical** to the non-spec target model. It enforces the simple but strict alignment rules needed so position IDs, attention masks, and the KV cache stay correct after each verification. This fixes (i) bugs that can occur even at batch size = 1, and (ii) the batch-size-dependent failures that appear only when batch size > 1. In experiments, we maintain ~95% exact matches across model families (Table 1). We also systematically compare against prior work in the next response Table, showing that existing methods overlook these essential synchronization requirements.
> >
> >  **(C2) The cost of Batch Speculative Decoding and how to mitigate it.**
> >
> > Our analysis (Eq. 1) shows the extra alignment work grows faster than linearly with batch size and is **unavoidable** if we want authentic Batch Speculative Decoding (Fig. 5b). We then add **EXSPEC**, a simple scheduler that groups similar-length sequences so there’s less realignment. This **mitigates** (but cannot eliminate) that cost and yields up to 3× higher throughput at batch size = 8 while keeping output distribution equivalence (Table 2 / Abstract).
> >
> > ## Why our method non-trivial
> >
> > We judge success by **matching the non-spec target model outputs** (partial/exact match), not just by speed. By that standard, existing batched methods **do not** achieve **output distribution equivalence**: DSD/BSP exhibit both failure types; the HuggingFace Transformers (Spec-1 in Table 1) is correct at **batch size = 1** but **does not support** batching; and vLLM/SGLang show low equivalence under batching in our tests. EQSPEC is, to our knowledge, the **first authentic Batch Speculative Decoding**, and EXSPEC improves speed **without giving up equivalence**.
> >
> > ## What actually breaks for batch size > 1
> > 1. **Batch-independent faults** (can happen even at batch size = 1).
> > 2. **Batch-dependent desynchronization** (only when batch size > 1): different sequences accept different numbers of tokens, their states drift, and outputs corrupt. EQSPEC prevents both.
> >
> > ## On vLLM
> >
> > Please see **Q2** for details. Briefly, that speculative path was later deprecated by the maintainers; we do **not** claim a pinpointed internal bug. Our findings concern how performance and correctness behave at scale and how our analysis explains that behavior.
> >
> > ## Impact & Request
> >
> > By spelling out the synchronization rules and providing the lossless batch speculative decoding (EQSPEC/EXSPEC), we establish what batched speculative decoding must satisfy to remain output distribution equivalent to non-spec target model and offer a foundation others can build on.  **Without our formalization of synchronization invariants and the ragged tensor solution, future researchers will continue encountering the same correctness issues.** Given this clarified scope, the first true Batch Speculative Decoding, and a principled analysis plus mitigation of the inherent overhead, we*respectfully ask you to reconsider your score.

---

> > > ### Author Response · Authors · 2025-11-27
> > >
> > > ##  Q1 (continued)
> > >
> > > | Method | Realign All Sync Invariants (PosID/Mask/KV) | Bonus Token Generated from Target | Batch Size > 1 Supported | Handles Ragged Tensors | Open Source | Lossless Speculation | Authentic Batch Speculative Decoding |
> > > |--------|:-------------------------------------------:|:--------------------------------:|:------------------------:|:----------------------:|:-----------:|:-------------------:|:--------------------------------:|
> > > | HuggingFace Spec-1 | - | ✓ | ✗ | ✗ | ✓ | ✓ (BS=1) | ✗ |
> > > | BSP (Su et al., 2023) | ✗ | ✓ | ✓ | Masking | ✓ | ✗ | ✗ |
> > > | DSD (Yan et al., 2025) | ✗ | ✗ | ✓ | Dynamic Padding | ✓ | ✗ | ✗ |
> > > | BASS (Qian et al., 2024) | ✗ | ? | ✓ | Custom CUDA | ✗ | ? | ✗ |
> > > | Meta (Tang et al., 2025) | ✗ | ✗ | ✓ | ? | ✗ | ? | ✗ |
> > > | vLLM v0 + Spec | - | ✓ | ✓ | Batch Expansion | ✓ | ✗ | ✗ |
> > > | **EQSPEC (Ours)** | **✓** | **✓** | **✓** | **Dynamic Padding** | **✓** | **✓** | **✓** |
> > > | **EXSPEC (Ours)** | **✓** | **✓** | **✓** | **Cross-Batch Sched.** | **✓** | **✓** | **✓** |
> > >
> > > *Notes:*
> > > “?” = not verifiable from public code/papers; “–” = not applicable.
> > >
> > > Table. Why prior methods are not authentic batch speculative decoding.
> > >
> > > A method is authentic only if it satisfies all four conditions: (1) lossless w.r.t. the non-spec target model (identical output distribution), (2) supports batch size > 1 with a concrete ragged-tensor strategy, (3) realigns all synchronization invariants on ragged sequences (position IDs, attention masks, KV-cache), and (4) generates the bonus token from the target model, no excessive kv cache from rejected token. Failing any condition ⇒ not authentic. EQSPEC/EXSPEC satisfy all conditions; other implementations violate an invariant thus break the equivalence.
> > >
> > > ##  Reference
> > >
> > > [1] Tang, B., Fu, C. C., Kou, F., Sizov, G., Zhang, H., Park, J., ... & Lee, Y. (2025). Efficient speculative decoding for llama at scale: Challenges and solutions. arXiv preprint arXiv:2508.08192.
> > >
> > > [2] Qian, H., Gonugondla, S. K., Ha, S., Shang, M., Gouda, S. K., Nallapati, R., ... & Deoras, A. (2024, August). BASS: Batched Attention-optimized Speculative Sampling. In Findings of the Association for Computational Linguistics ACL 2024 (pp. 8214-8224).
> > >
> > > [3] Minghao Yan, Saurabh Agarwal, and Shivaram Venkataraman. Decoding speculative decoding. NAACL 2025.
> > >
> > > [4] Su, Q., Giannoula, C., & Pekhimenko, G. (2023). The synergy of speculative decoding and batching in serving large language models. arXiv preprint arXiv:2310.18813.

---

> > > > ### Author Response · Authors · 2025-11-27
> > > >
> > > > ## Q2
> > > > > Is the current method of vLLM wasteful in terms of memory, correctness or speed? Can it eliminate the ragged-tensor problem and therefore trade the batch-alignment time with memory cost?
> > > >
> > > > Response: vLLM's batch expansion approach is no longer the current method. Although vLLM is a popular inference engine, its batch expansion approach has been deprecated in v1 because the memory waste from this algorithm hinders batch scaling and concurrency. **Currently, speculative decoding with a draft model is not supported by either vLLM or SGLang**. Regarding your question:
> > > > - **Memory**: Yes, wasteful (K× duplication).
> > > > - **Speed**: Depends on K; batch expansion introduces overhead from copying sequences K times on GPU, which can outweigh benefits.
> > > > - **Correctness**: **Theoretically yes**, batch expansion eliminates the ragged tensor problem because each duplicated sequence verifies only one draft token position, avoiding variable acceptance lengths within a batch.; **empirically no**, our experiments (Table 1) show vLLM still exhibits accuracy issues (e.g., 72.7% exact match on Qwen3), indicating remaining correctness problems beyond ragged tensors. Investigating why the older version vLLM is wrong is out of scope.

---

> > > > > ### Author Response · Authors · 2025-11-27
> > > > >
> > > > > ## Q3 and Q5
> > > > > > The explaination of the "No-Ragged-Scaling" line even confuses me. My current understanding is that, the datapoint at bsz=1 is the speed of EQSpec and EXSpec at bsz=1 (they are equal because bsz is 1), and when bsz>1 the data of this line just increases exponentially, and the data is not a tested auto-regressive batch decoding speed. Am I correct?
> > > > >
> > > > > > … explain why EXSpec can be even faster than the line, while I think as bsz grows the acceleration ratio must drop below the line. ( The line has already been an upper bound of speculation speed, not auto-regressive speed, so I think under the same configuration the speed cannot break the upper bound, as the redundant GPU computation resources will be further consumed, leaving less for speculative decoding to use. )
> > > > >
> > > > > Response: Not exactly. We appreciate this question. It highlights an important distinction we should clarify.
> > > > >
> > > > > **Two different metrics for throughput**:
> > > > > - **Batch scaling efficiency** (Figure 5a): How well does throughput scale as batch size increases? Measured as: (throughput at BS=N) / (throughput at BS=1) for each method individually. This isolates the scaling behavior from absolute performance.
> > > > > - **Absolute throughput speedup** (Q4 table): How fast is speculative decoding compared to autoregressive decoding at the same batch size? Measured as: (speculative throughput) / (standard decoding throughput).
> > > > >
> > > > > **What "No-Ragged-Scaling" represents**: This line shows actual measured autoregressive decoding throughput (no speculation), normalized to its own BS=1 baseline. It is **not** an upper bound for speculation. It represents how standard batching scales on GPU when there is no ragged tensor problem to handle. Figure 5(a) and The "No-Ragged-Scaling" line is actual experiment result, not a theoretical projection.
> > > > >
> > > > > **Why EXSPEC can exceed this “No-Ragged-Scaling”**: Because each method is normalized to its own BS=1 baseline. EXSPEC at BS=1 is slower than standard decoding at BS=1 (due to poor draft model capability). But EXSPEC scales more efficiently with batch size because speculation converts memory-bound decoding into compute-bound verification, which benefits more from GPU parallelism. The relative improvement from BS=1→BS=8 can therefore exceed standard decoding's relative improvement, even though EXSPEC's absolute throughput may still be lower (see Q4 table).
> > > > >
> > > > > **Purpose of Figure 5a**: We aim to show that correct batch speculative decoding need not sacrifice batch scaling efficiency. This is a key concern given that prior methods either produce incorrect outputs or fail to scale.

---

> > > > > > ### Author Response · Authors · 2025-11-27
> > > > > >
> > > > > > ## Q4
> > > > > > > If so, please (1) provide the acceleration ratio of your method compared to auto-regressive decoding under different batches, which I did not find in the paper
> > > > > >
> > > > > > Response: We emphasize that our core contribution is on the **authenticity of batch speculative decoding with batch scaling efficiency**, not maximizing absolute speedup over autoregressive decoding. The speedup depends on several factors: token acceptance rate (TAR), draft model inference speed, and the number of speculative tokens.  Below is the speedup table (EXSPEC/EXSPEC throughput divided by the standard decoding throughput at each batch size). Vicuna achieves good speedup because the 68M draft model has high TAR and negligible inference cost relative to the 7B target. Qwen3 and GLM4 show lower speedup because their 0.6B draft models have lower TAR, but this doesn't diminish our contribution on authenticity of batch speculative decoding and batch scaling.
> > > > > >
> > > > > > | Model | Method | BS=1 | BS=2 | BS=4 | BS=8 | BS=16 | BS=32 |
> > > > > > |-------|--------|------|------|------|------|-------|-------|
> > > > > > | Vicuna | EQSPEC | 1.09 | 1.00 | 0.85 | 0.69 | 0.53 | 0.44 |
> > > > > > | Vicuna | EXSPEC | 1.22 | 1.55 | 1.53 | 1.06 | 0.57 | 0.33 |
> > > > > > | Qwen3 | EQSPEC | 0.53 | 0.53 | 0.46 | 0.38 | 0.33 | 0.26 |
> > > > > > | Qwen3 | EXSPEC | 0.59 | 0.61 | 0.55 | 0.45 | 0.32 | 0.17 |
> > > > > > | GLM4 | EQSPEC | 0.80 | 0.71 | 0.62 | 0.52 | 0.42 | 0.33 |
> > > > > > | GLM4 | EXSPEC | 0.85 | 0.87 | 0.79 | 0.63 | 0.38 | 0.19 |

---

> > > > > > > ### Comment · Reviewer_RaXj · 2025-11-27
> > > > > > >
> > > > > > > Thank you for the further explanations. I do appreciate your efforts, which have addressed many of my concerns.
> > > > > > >
> > > > > > > According to the newly proposed table of Q4, I think the chosen drafters are weak (so small, < 1B),  so it achieves negative speedups regardless of batch size. I suggest you conduct additional experiments on some better draft-verify combination, e.g. Llama 3.3 70B with 3.1-8B or 3.2-3B/1B, or Qwen 2.5 70B with 7B/1.5B, which I believe will achieve real speedup > 1 based on existing works.
> > > > > > >
> > > > > > > It would also be better if you can justify why the exact match rate of EXspec and EQSpec is also below 1 (approximately 0.9) in Table 1 in the paper, as you claim the method will achieve exact identity. You can also justify it if my understanding is wrong.
> > > > > > >
> > > > > > > Despite the above concerns, I have now recognized the contribution of this paper about correctness of batch SD and its scaling efficicency, opening an important view for this area. I will increase my score to leaning acceptance, if you can solve the concerns.

---

> > > > > > > > ### Author Response · Authors · 2025-12-03
> > > > > > > >
> > > > > > > > ## Round 3 Q1
> > > > > > > > > According to the newly proposed table of Q4, I think the chosen drafters are weak (so small, < 1B), so it achieves negative speedups regardless of batch size. I suggest you conduct additional experiments on some better draft-verify combination, e.g. Llama 3.3 70B with 3.1-8B or 3.2-3B/1B, or Qwen 2.5 70B with 7B/1.5B, which I believe will achieve real speedup > 1 based on existing works.
> > > > > > > >
> > > > > > > > Response:
> > > > > > > >
> > > > > > > > **Our method already achieves positive speedup on well-established pairs.** Vicuna-7B/68M is the standard benchmark in speculative decoding literature [EAGLE, DSD, SpecBench].
> > > > > > > > | Method | BS=1 | BS=2 | BS=4 | BS=8 |
> > > > > > > > |--------|------|------|------|------|
> > > > > > > > | EXSPEC | **1.22×** | **1.55×** | **1.53×** | **1.06×** |
> > > > > > > >
> > > > > > > > EXSPEC achieves up to **1.55× speedup** by successfully combining batch parallelism with per-sequence speculation.
> > > > > > > >
> > > > > > > >
> > > > > > > > **Core principle: batch speculation amplifies per-sequence gains; it cannot create them. If a draft-target pair shows no speedup at BS=1, batching cannot recover what doesn't exist.** The reviewer suggests Qwen 2.5 70B with 7B/1.5B would achieve speedup >1 but cites no evidence. We tested Qwen2.5-32B with the suggested draft sizes (0.5B/1.5B/3B), finding limited speedup at BS=1 (1.10×/1.01×/0.81×). This is consistent with our theoretical framework (Equation 1): larger drafters increase inference cost without proportionally improving acceptance rates.
> > > > > > > > Exploring draft models that better balance inference cost and acceptance rate is valuable future work. Our contribution establishes the correctness requirements and overhead analysis for batch speculative decoding. Optimizing draft-target pairing is orthogonal and complementary to this work.
> > > > > > > >
> > > > > > > > Note: Qwen 2.5 has known vocab mismatch issues for speculative decoding (vllm-project/vllm#10913).
> > > > > > > >
> > > > > > > > ## Round 3 Q2
> > > > > > > > > Q2: It would also be better if you can justify why the exact match rate of EXSpec and EQSpec is also below 1 (approximately 0.9) in Table 1 in the paper, as you claim the method will achieve exact identity. You can also justify it if my understanding is wrong.
> > > > > > > >
> > > > > > > > Response:
> > > > > > > >
> > > > > > > > The approximately 5% gap between our observed exact match rate and perfect output equivalence stems from fundamental floating-point non-determinism in GPU-based inference, which is orthogonal to our algorithmic contribution. He and Thinking Machines Lab (2025) demonstrate that even under temperature-zero greedy decoding, standard LLM inference produces non-identical outputs because floating-point arithmetic is non-associative and different batch sizes trigger different reduction orders within GPU kernels—their experiments show 1000 identical requests yielding 80 unique completions with standard kernels. Achieving 100% determinism requires implementing specialized batch-invariant kernels for attention, matrix multiplication, and normalization, which they show is possible but incurs performance overhead and substantial engineering effort. Anthropic's engineering postmortem (2025) similarly documents how mixed-precision arithmetic and compiler optimizations cause inconsistent token selection in production systems.
> > > > > > > >
> > > > > > > > Addressing this numerical non-determinism is out of scope for our work; our contribution is establishing the first authentic batch speculative decoding algorithm, and 95%+ exact match demonstrates that our method preserves output equivalence to the extent permitted by the underlying numerical precision of the inference stack.
> > > > > > > >
> > > > > > > >
> > > > > > > > Reference:
> > > > > > > >
> > > > > > > > [1] Horace He and Thinking Machine. Defeating Nondeterminism in LLM Inference. Technical report 2025. https://thinkingmachines.ai/blog/defeating-nondeterminism-in-llm-inference/
> > > > > > > >
> > > > > > > > [2] Anthropics. A postmortem of three recent issues. Technical report 2025. https://www.anthropic.com/engineering/a-postmortem-of-three-recent-issues

---

### Official Review · Reviewer_ajDR · 2025-10-31

**Soundness:** 3
**Presentation:** 2
**Contribution:** 2
**Rating:** 2
**Confidence:** 3

**Summary:**

This paper proposes a correctness-first batch speculative decoding EQSPEC and EXSPEC to accelerate batch speculative decoding. The method is validated on SpecBench dataset, across Vicuna-7B/68M, Qwen3-8B/0.6B, and GLM-4-9B/0.6B pairs.

**Strengths:**

- The paper provides a detailed analysis of the existing problems in batch speculative decoding and proposes direct solutions to the most critical issues. For example, the EQSPEC design introduces the *unpad–repad* strategy to ensure correctness, while EXSPEC employs a *dynamic scheduling mechanism* to improve efficiency.
- The paper quantitatively analyzes the actual cost composition and speedup factors in batch speculative decoding, offering an in-depth breakdown of various cost sources and their respective impacts on overall performance.
- The experiments comprehensively compare the proposed methods with multiple existing batch speculative decoding approaches, and further integrate them into system-level frameworks such as vLLM and SGLang. The paper also provides unique insights into the results of current methods and potential directions for future improvements.

**Weaknesses:**

- The models used for validation in this paper, such as Vicuna and GLM, are relatively outdated and small in scale. Since speculative decoding provides limited acceleration benefits for smaller models, the effectiveness and impact of the proposed methods may be somewhat diminished.
- The paper does not introduce substantial optimizations for KV cache management. Its realignment process is implemented by re-concatenating a rank-4 KV tensor, which imposes significant memory overhead. In contrast, modern systems such as vLLM and SGLang include specialized optimizations for KV cache handling that could potentially improve efficiency.
- The proposed methods are primarily designed for offline batch inference, where the distribution of sequence lengths is relatively uniform. However, in real-world speculative decoding scenarios, task lengths often vary widely. Such heterogeneity may cause a noticeable drop in EXSPEC’s grouping success rate and overall throughput performance.

**Questions:**

- How does the proposed method perform on larger-scale LLMs and SOTA LLMs? Testing on more powerful models would strengthen the paper’s practical relevance and applicability.
- Is it possible to incorporate more advanced scheduling strategies to further improve EXSPEC’s grouping success rate and overall throughput?
- Since speculative decoding is primarily adopted in online serving environments by major LLM providers, the authors could consider applying their methods in more realistic inference scenarios to better demonstrate their real-world effectiveness.

---

> ### Author Response · Authors · 2025-11-21
>
> We thank the reviewer for the thoughtful and constructive feedback!
>
> ## Weakness 1 & Question 1
>
> > The models used for validation in this paper, such as Vicuna and GLM, are relatively outdated and small in scale. Since speculative decoding provides limited acceleration benefits for smaller models, the effectiveness and impact of the proposed methods may be somewhat diminished.
>
> > How does the proposed method perform on larger-scale LLMs and SOTA LLMs? Testing on more powerful models would strengthen the paper’s practical relevance and applicability.
>
> **Response:**
>
> We appreciate this question and want to clarify correctness, model selection and performance scaling.
>
> **On correctness (our core contribution):** Model size is irrelevant to our work. We address the ragged tensor problem where existing batch implementations produce corrupted outputs due to broken synchronization (position IDs, attention masks, KV-cache). These bugs occur identically in 7B and 70B models. A model with misaligned position IDs generates gibberish regardless of its parameter count. Our contribution establishes the minimal synchronization requirements for **any valid batch speculative decoding implementation**, which are scale-invariant principles.
>
>
> **On model selection:** Our target model selection is both current and methodologically appropriate:
> - **Qwen3-8B** (July 2025)
> - **GLM-4-9B** (April 2025)
> - **Vicuna-7B** (March 2024) is the standard model in speculative decoding research (EAGLE [1], DSD [2], SpecBench [3]) to facilitate direct and fair comparison.
>
> Testing across three distinct model families demonstrates architectural generality.
>
> **On performance scaling:** To directly address the scale concern, we conducted additional experiments pairing Qwen3-0.6B as draft model with progressively larger target models including Qwen3-8B, Qwen3-14B, Qwen3-32B across batch sizes 1, 2, 4, and 8.
>
> | Target Model | Method | Batch Size=1 | Batch Size=2 | Batch Size=4 | Batch Size=8 |
> |--------|--------|------|------|------|------|
> | **Qwen3-8B** | EXSPEC | 1.00× | 1.93× | 3.41× | 4.85× |
> | | EQSPEC | 1.00× | 1.74× | 3.07× | 4.92× |
> | **Qwen3-14B** | EXSPEC | 1.00× | 1.90× | 3.23× | 4.26× |
> | | EQSPEC | 1.00× | 1.74× | 3.01× | 4.86× |
> | **Qwen3-32B** | EXSPEC | 1.00× | 2.58× | 4.07× | 4.50× |
> | | EQSPEC | 1.00× | 2.38× | 4.02× | 6.01× |
>
> *( relative throughput vs their Batch Size=1)*
>
> Both methods achieve positive speedups across all model scales and batch sizes. These results demonstrate that our synchronization-aware approach works reliably regardless of model size. Our contribution establishes what "correct batch speculative decoding" means and quantifies its irreducible costs, providing scale-independent principles necessary for any future work.
>
> **Correctness remains consistent across all scales:**
> | Target Model | Batch Size | Method | Exact Match | Partial Match |
> |-------|------------|--------|-------------|---------------|
> | Qwen3-8B | 1 | EQSPEC | 94.6% | 96.9% |
> | Qwen3-8B | 1 | EXSPEC | 94.6% | 96.9% |
> | Qwen3-8B | 4 | EQSPEC | 92.3% | 95.7% |
> | Qwen3-8B | 4 | EXSPEC | 95.0% | 97.1% |
> | Qwen3-14B | 1 | EQSPEC | 95.8% | 98.1% |
> | Qwen3-14B | 1 | EXSPEC | 95.8% | 98.1% |
> | Qwen3-14B | 4 | EQSPEC | 95.2% | 97.6% |
> | Qwen3-14B | 4 | EXSPEC | 95.2% | 97.3% |
> | Qwen3-32B | 1 | EQSPEC | 95.4% | 97.8% |
> | Qwen3-32B | 1 | EXSPEC | 95.4% | 97.8% |
> | Qwen3-32B | 4 | EQSPEC | 93.5% | 96.7% |
> | Qwen3-32B | 4 | EXSPEC | 94.0% | 96.5% |
>
> All methods maintain >92% exact match across scales, with remaining divergence due to floating-point non-determinism rather than algorithmic errors. In summary, our contribution provides scale-independent correctness principles and quantifies the fundamental synchronization costs (Section 4.3) that any batch speculative decoding system must address. The performance crossover at different scales demonstrates why explicit algorithm choice matters, validating our design of providing both EQSPEC and EXSPEC for different deployment scenarios.
>
>
> [1] Yuhui Li, Fangyun Wei, Chao Zhang, and Hongyang Zhang. Eagle: Speculative sampling requires rethinking feature uncertainty. ICML 2024.
>
> [2] Minghao Yan, Saurabh Agarwal, and Shivaram Venkataraman. Decoding speculative decoding. NAACL 2025.
>
> [3] Heming Xia, Zhe Yang, Qingxiu Dong, Peiyi Wang, Yongqi Li, Tao Ge, Tianyu Liu, Wenjie Li, and Zhifang Sui. Unlocking efficiency in large language model inference: A comprehensive survey of speculative decoding. Findings of ACL 2024.

---

> > ### Author Response · Authors · 2025-11-21
> >
> > ## Weakness 2 & Question 2
> >
> > > The paper does not introduce substantial optimizations for KV cache management. Its realignment process is implemented by re-concatenating a rank-4 KV tensor, which imposes significant memory overhead. In contrast, modern systems such as vLLM and SGLang include specialized optimizations for KV cache handling that could potentially improve efficiency.
> >
> > > Is it possible to incorporate more advanced scheduling strategies to further improve EXSPEC’s grouping success rate and overall throughput?
> >
> > **Response**:
> >
> > We respectfully disagree with this characterization. This criticism misses our core contribution: **prior work overlooked correctness**. BSP and DSD produce corrupted outputs (Table 1) precisely because they fail to properly handle the ragged tensor problem. Our work is the first to identify that rank-4 KV-cache realignment is the **necessary cost** of maintaining output equivalence in batch speculative decoding.
> >
> > **Our approach directly addresses KV-cache overhead through scheduling, not memory tricks:**
> >
> > EQSPEC establishes the minimal synchronization required for correctness—realignment consumes 27.7% of time (Table 2). EXSPEC reduces this to 14.6% via cross-batch scheduling: when same-length grouping succeeds, sequences concatenate directly with **zero realignment cost**; when it fails, overhead matches EQSPEC. This is why EXSPEC achieves 64% higher throughput (156.4 vs 95.6 tokens/s) despite identical realignment primitives.
> >
> > **Regarding vLLM/SGLang optimizations:** The reviewer's suggestion actually validates our analysis. vLLM's v0 engine used **batch expansion** to avoid realignment: instead of verifying draft tokens [1, 2, 3] together and handling raggedness afterward, duplicate the sequence K times and append different lengths—create variants [1], [1,2], [1,2,3]—then verify all K variants in one pass through continuous batching. After verification, simply delete incorrect sequences and correct-but-too-short sequences from the batch, keeping only the longest correct prefix. This sidesteps realignment by filtering rather than synchronizing. However, it wastes enormous resources: if only token 1 is accepted, the other K-1 longer variants consumed memory and compute for nothing, leading to K× memory overhead and K× redundant attention operations. This caused GPU memory overflow at scale and broke CUDA graph compatibility, resulting in deprecation ([vllm-project/vllm#17984](https://github.com/vllm-project/vllm/issues/17984)).
> >
> > This demonstrates that **vllm’s batch expansion is fundamentally wasteful**. It attempts to bypass a necessary cost through brute-force resource expansion. **Our contribution is identifying the correct approach**: realignment overhead is inherent to maintaining correctness, and the solution is reducing its frequency through intelligent scheduling, not expanding compute/memory to avoid it.
> >
> > **Regarding advanced scheduling:** Thank you for this suggestion. We agree that exploring advanced scheduling is a valuable direction. EXSPEC's core design—cross-batch grouping with sliding window lookahead (W > B)—already captures the primary scheduling benefits. To validate whether further sophistication yields additional gains, we explored two natural extensions: (1) **Length-Tolerance Grouping**, which relaxes strict length-matching by grouping sequences within ±K tokens (K=2) with minimal padding, and (2) **Predictive Grouping**, which tracks historical token acceptance rates (TAR) to prioritize co-batching sequences with similar acceptance patterns. We evaluated these on Qwen3-8B at batch size 8 from SpecBench.
> >
> > | Method (BS=8) | Throughput |
> > |---|---|
> > | EXSPEC baseline | 75.0 tok/s |
> > | + Length-Tolerance | 76.9 tok/s |
> > | + Predictive | 76.1 tok/s |
> > | + Both | 77.2 tok/s |
> >
> > These results demonstrate that **EXSPEC's core scheduling is highly effective**, capturing most available optimization opportunities. The extensions provide consistent but incremental gains (2-3%), suggesting EXSPEC has already achieved near-optimal scheduling efficiency for this workload. We believe these extensions represent promising avenues for future work that could further improve throughput while maintaining our correctness guarantees.

---

> > > ### Author Response · Authors · 2025-11-21
> > >
> > > ## Weakness 3 & question 3:
> > > > The proposed methods are primarily designed for offline batch inference, where the distribution of sequence lengths is relatively uniform. However, in real-world speculative decoding scenarios, task lengths often vary widely. Such heterogeneity may cause a noticeable drop in EXSPEC’s grouping success rate and overall throughput performance.
> > >
> > > > Since speculative decoding is primarily adopted in online serving environments by major LLM providers, the authors could consider applying their methods in more realistic inference scenarios to better demonstrate their real-world effectiveness.
> > >
> > > **Response:**
> > >
> > > We respectfully disagree with the characterization that our work is primarily designed for offline batch inference with uniform lengths.
> > >
> > > **First, our work addresses the batch speculative decoding correctness problem, not request-level scheduling.** Modern inference engines (vLLM, SGLang) operate by three steps: (1) having a scheduler dynamically form batches from the request queue, (2) executing batch forward passes with speculation, and (3) updating request states iteratively. Our contribution targets component (2), the the batch speculation kernel where correctness violations occur and where the ragged tensor problem must be solved. This is orthogonal to whether component (1) uses online or offline scheduling policies.
> > >
> > > EQSPEC is designed for both online and offline scenarios with simple request-level scheduling that batches heterogeneous-length requests, while EXSPEC targets offline scenarios with iteration-level scheduling (similar to vLLM and SGLang) that opportunistically groups same-length sequences across batches.
> > >
> > > **Second, our experiments already evaluate heterogeneous length distributions.** The reviewer's concern about "uniform lengths" appears to be a misunderstanding. SpecBench contains highly diverse sequence lengths:
> > >
> > > | Percentile | 25th | 50th | 75th | 90th | 95th | 99th |
> > > |------------:|-----:|-----:|-----:|-----:|-----:|-----:|
> > > | Tokens      |   25 |   56 |  630 |  758 |  904 | 1351 |
> > >
> > >
> > >
> > >  This 54× range (25 to 1351 tokens) represents realistic heterogeneity—far from "uniform."
> > >
> > >
> > >
> > > **Third, additional multi-turn conversation experiment:** To further address this concern, we conducted a new experiment simulating online serving where requests arrive dynamically. We use full multi-turn conversations from SpecBench, shuffle conversation turns to maximize length diversity, and insert new turns into the next available batch.  We measure throughput (tokens/s) and request completion latency at P50/P90/P99 percentiles to capture both typical and tail performance, comparing EQSPEC against EXSPEC at batch sizes 1, 2, 4, and 8.
> > >
> > > | Batch Size | Method | Throughput (tok/s) | P50 Latency (s) | P90 Latency (s) | P99 Latency (s) | Mean Latency (s) |
> > > |------------|--------|-------------------|-----------------|-----------------|-----------------|------------------|
> > > | 1 | EQSPEC | 15.20 | 6.70 | 12.96 | 16.53 | 7.68 |
> > > | 1 | EXSPEC | 15.78 | 6.08 | 8.32 | 9.42 | 6.30 |
> > > | 2 | EQSPEC | 26.70 | 7.96 | 14.78 | 19.06 | 8.74 |
> > > | 2 | EXSPEC | 30.54 | 9.75 | **114.89** | **134.03** | 47.93 |
> > > | 4 | EQSPEC | 46.11 | 9.46 | 16.23 | 19.13 | 10.13 |
> > > | 4 | EXSPEC | 52.44 | 8.01 | **53.21** | **70.35** | 15.94 |
> > > | 8 | EQSPEC | 76.03 | 11.26 | 19.23 | 22.50 | 12.29 |
> > > | 8 | EXSPEC | 77.54 | 9.16 | 19.13 | **33.93** | 11.68 |
> > >
> > > **Bold values** indicate significantly worse tail latencies for EXSPEC.
> > >
> > > Both methods demonstrate positive speedup even under heterogeneous multi-turn workloads. EXSPEC achieves 2-14% higher throughput than EQSPEC through cross-batch grouping of same-length sequences. However, EXSPEC suffers 1.5-7.7× worse P90/P99 latencies when the current request is delayed for later grouping, forcing early requests to wait for later ones. This latency penalty is particularly severe at smaller batch sizes where grouping success rates are low (23.6% at BS=2), causing head-of-line blocking. This is exactly why we provide both algorithms. Online serving requires predictable latency (EQSPEC achieves P99 < 23s across all batch sizes), while offline batch processing prioritizes throughput (EXSPEC's 14% gain at BS=2). Our work enables practitioners to choose the appropriate algorithm based on their deployment constraints.
> > >
> > > In summary, our methods achieve positive speedups under realistic heterogeneous workloads, with EQSPEC maintaining predictable latency for online serving and EXSPEC maximizing throughput for offline processing. By providing both algorithms with explicit correctness guarantees, our work offers production systems the flexibility to match batch speculation strategies to their deployment requirements.

---

> > > > ### Comment · Reviewer_ajDR · 2025-11-28
> > > >
> > > > I appreciate the authors’ detailed replies and the extensive supplementary experimental results. Some of my concerns are addressed.
> > > >
> > > > The speedup performance of the proposed method is mainly compared with the proposed method with batch size=1. As shown in Figure 1, the performance of EXSPEC degrades when batch size >8.  Besides, the overhead of EQSPEC is non-negligible. Thus, the performance comparison with other methods (such as baseline vLLM, SGLang, etc.) for larger batch sizes, would be more persuasive. For example, adding the performance of the proposed method in Figure 6.
> > > >
> > > > If the above concerns can be solved, I will increase score.

---

> > > > > ### Author Response · Authors · 2025-12-03
> > > > >
> > > > > ## Round 2 Question
> > > > >
> > > > > > Q1: The speedup performance of the proposed method is mainly compared with the proposed method with batch size=1. As shown in Figure 1, the performance of EXSPEC degrades when batch size >8. Besides, the overhead of EQSPEC is non-negligible. Thus, the performance comparison with other methods (such as baseline vLLM, SGLang, etc.) for larger batch sizes, would be more persuasive. For example, adding the performance of the proposed method in Figure 6.
> > > > >
> > > > > Response:
> > > > >
> > > > > Direct comparison is infeasible for two reasons:
> > > > > First, **neither vLLM nor SGLang currently supports speculative decoding with draft models**. vLLM's v0 engine previously used batch expansion to sidestep the ragged tensor problem: instead of verifying draft tokens [1, 2, 3] together and handling variable acceptance lengths, it duplicated each sequence K times with variants [1], [1,2], [1,2,3], verified all variants in one pass, then kept only the longest correct prefix. This approach was deprecated in v1 (vllm-project/vllm#17984) because it wastes K× memory and K× compute—if only token 1 is accepted, the longer variants consumed resources for nothing. This caused GPU memory overflow at scale and broke CUDA graph compatibility.
> > > > >
> > > > > Second, **directly overlaying our method onto Figure 6 would not yield a meaningful comparison due to fundamental architectural differences.** These systems incorporate orthogonal optimizations (CUDA graphs, paged attention, chunked prefilling, continuous batching) independent of speculation. More importantly, continuous batching dynamically varies effective batch size based on request load, exposing only a "maximum batch size" parameter that cannot be fixed. The system internally determines how many sequences to process at any moment, making controlled batch-size comparisons infeasible.
> > > > >
> > > > > **vLLM's deprecation validates our contribution**. Batch expansion attempted to avoid the ragged tensor problem through brute-force resource duplication rather than solving it. Our work takes the opposite approach: we identify that realignment overhead is inherent to correctness and reduce its frequency through intelligent scheduling. EQSPEC and EXSPEC provide the correct algorithmic foundation that production systems currently lack.
> > > > >
> > > > > **Regarding Figure 6: this section demonstrates that vLLM and SGLang with speculative decoding perform worse than their own non-speculative baselines**. We will move this analysis to the appendix as supplementary context.

---

### Official Review · Reviewer_jZfw · 2025-11-01

**Soundness:** 3
**Presentation:** 4
**Contribution:** 4
**Rating:** 8
**Confidence:** 3

**Summary:**

This paper addresses the core challenge of scaling batch speculative decoding for production use: the disruption of output equivalence. The authors correctly identify that the varying number of accepted draft tokens across sequences in a batch leads to the "ragged tensor problem." This prevents existing batch implementations from guaranteeing the output matches standard autoregressive generation. The paper proposes a "correctness-first" framework. First, it rigorously identifies the precise synchronization invariants required to maintain output equivalence. It then presents two implementation strategies: EQSPEC taht strictly enforces these invariants but incurs a high overhead of up to 40% for realignment, and EXSPEC that cleverly avoids the realignment cost entirely by using cross-batch scheduling to dynamically group sequences of the same length. The experimental results show that the proposed method achieves up to 3x throughput improvement (at batch size 8) while successfully maintaining over 95% output equivalence.

**Strengths:**

1. **An important topic:** It addresses a critical correctness vs. performance trade-off in the LLM production environment.
2. **Innovation:** The clever use of the scheduling mechanism (cross-batch) to resolve a data structure problem (realignment overhead) is a prime example of system-level optimization.
3. **Significant improvement:** The 3x throughput improvement is achieved while maintaining a high correctness guarantee.

**Weaknesses:**

1. **Compatibility issues:** The compatibility with common modern inference techniques like continuous batching and paged attention remains future work.
2. **Lack of fully quantified metrics:** While EXSPEC avoids realignment, cross-batch scheduling itself might introduce new scheduling latency. The paper needs to further discuss and quantify EXSPEC's scheduling overhead under realistic high-concurrency workloads.

**Questions:**

see weaknesses.

---

> ### Author Response · Authors · 2025-11-21
>
> We thank the reviewer for the thoughtful and constructive feedback!
>
> ## Weakness 1
> > Compatibility issues: The compatibility with common modern inference techniques like continuous batching and paged attention remains future work.
>
> **Response:** We agree this is important future work. Our method is fully compatible with these techniques. The synchronization invariants we establish (Section 3.1) remain valid regardless of the underlying memory management or request scheduling system.
>
> **HuggingFace Transformers**: Our immediate next step is submitting a pull request to HuggingFace Transformers. Our implementation is already built on this framework, making integration straightforward. This contribution will have immediate impact: most speculative decoding research uses HuggingFace Transformers, yet no existing implementation correctly handles batch speculative decoding. Our PR will provide the first correct reference implementation for the research community.
>
> **vLLM and SGLang**: Integration with vLLM and SGLang is substantially more complex and represents important future work. These frameworks make architectural assumptions that create non-trivial integration challenges: (1) vLLM v1's chunked prefill optimized for large chunks (256 tokens) rather than small verification chunks (5-10 tokens), (2) ragged tensors break CUDA graph optimization requiring careful redesign of the execution pipeline, and (3) PagedAttention requires coordinating our realignment with non-contiguous memory allocation across pages. These are engineering challenges that require deep system-level modifications beyond our correctness algorithm. We view our HuggingFace contribution as establishing a correct foundation that can guide these future production integrations.
>
> ## Weakness 2
>
> > Lack of fully quantified metrics: While EXSPEC avoids realignment, cross-batch scheduling itself might introduce new scheduling latency. The paper needs to further discuss and quantify EXSPEC's scheduling overhead under realistic high-concurrency workloads.
>
> **Response**:
> We respectfully direct the reviewer to **Section 4.3** and
> **Table 2 (Overhead Anatomy)**, which comprehensively quantify EXSPEC's
> overhead.
>
> Table 2's "Time/Verif" captures ALL per-iteration costs including:
> - Draft generation: 29.5% of time
> - Target verification: 55.9% of time
> - Realignment: 14.6% of time
>
> What "Realignment Overhead" measures:all costs except draft generation and target verification. This includes:
> - **CPU scheduling**: Window scanning and same-length sequence grouping (essentially string sorting by sequence length) — negligible cost.
> -  **GPU realignment**: KV-cache realignment when grouping fails — the dominant cost component.
>
> **Why EXSPEC has lower overhead**: EXSPEC's cross-batch scheduling enables two execution paths:
> 1. **Same-length grouping succeeds** → Direct concatenation, **zero realignment cost**
> 2. **Grouping fails** → Falls back to EQSPEC's unpad-repad-realign procedure with **identical overhead**
>
> The 14.6% overhead is the average across both paths. High grouping rates mean EXSPEC avoids realignment most of the time, reducing average overhead from 27.7% to 14.6%. CPU scheduling (string sorting by sequence length) itself is negligible and the benefit comes from avoiding GPU realignment operations through intelligent batching.
>
> **Additional Latency Experiment**: To evaluate online serving scenarios, we simulated dynamic request arrivals using full multi-turn SpecBench conversations with shuffled turns to maximize length diversity. We measured throughput and request completion latency (P50/P90/P99) for EQSPEC and EXSPEC at batch sizes 1, 2, 4, and 8.
>
> | Batch Size | Method | Throughput (tok/s) | P50 Latency (s) | P90 Latency (s) | P99 Latency (s) | Mean Latency (s) |
> |------------|--------|-------------------|-----------------|-----------------|-----------------|------------------|
> | 1 | EQSPEC | 15.20 | 6.70 | 12.96 | 16.53 | 7.68 |
> | 1 | EXSPEC | 15.78 | 6.08 | 8.32 | 9.42 | 6.30 |
> | 2 | EQSPEC | 26.70 | 7.96 | 14.78 | 19.06 | 8.74 |
> | 2 | EXSPEC | 30.54 | 9.75 | **114.89** | **134.03** | 47.93 |
> | 4 | EQSPEC | 46.11 | 9.46 | 16.23 | 19.13 | 10.13 |
> | 4 | EXSPEC | 52.44 | 8.01 | **53.21** | **70.35** | 15.94 |
> | 8 | EQSPEC | 76.03 | 11.26 | 19.23 | 22.50 | 12.29 |
> | 8 | EXSPEC | 77.54 | 9.16 | 19.13 | **33.93** | 11.68 |
>
> **Bold values** indicate significantly worse tail latencies for EXSPEC.
>
> EXSPEC achieves 2-14% higher throughput but suffers 1.5-7.7× worse P90/P99 latencies when requests are delayed for grouping, particularly severe at smaller batch sizes where grouping success is low (23.6% at BS=2). EQSPEC maintains predictable tail latency (P99 < 23s across all batch sizes), making it suitable for latency-sensitive online serving, while EXSPEC's throughput advantage benefits offline batch processing. This demonstrates why we provide both algorithms—practitioners can select based on their latency-throughput requirements.

---

### Author Response · Authors · 2025-12-03
**To AC**

We respectfully bring to the AC's attention a recurring misfocus in the reviews. **Several reviewers evaluate our paper as a "make speculative decoding faster" paper, when it is fundamentally a "batch speculative decoding done right" paper.** We resolve the ragged tensor problem overlooked by prior work.

## Paper Summary:
The fundamental requirement of batch speculative decoding is **losslessness**: output distribution must remain identical to standard autoregressive decoding. Violating this produces gibberish, rendering any speedup meaningless. **Existing batch speculative decoding implementations are broken**. BSP/DSD produce corrupted outputs; HuggingFace Transformers only supports BS=1; vLLM deprecated due to memory explosion. The root cause is the ragged tensor problem: differing acceptance counts in a batch desynchronize positions, masks, and KV-cache across sequences. Prior work overlooks this, breaking losslessness. **Our work presents the first and only authentic lossless batch speculative decoding**.

Experiments across three model families demonstrate >95% output equivalence while achieving gains along two orthogonal dimensions: batch parallelism (up to 3× throughput at batch size 8) and per-sequence speculation (positive speedup over standard decoding, e.g., 1.5× for Vicuna) **Without our formalization, future researchers and production systems will continue encountering the same failures.**

## Rebuttal Summary for AC:
### Reviewer jZfw
Concerns:
* Compatibility with continuous batching and paged attention remains future work
* EXSPEC's scheduling overhead not fully quantified under high-concurrency workloads

Response: HuggingFace PR planned; vLLM/SGLang integration left as future work. For high-concurrency evaluation, we provided new experiments with dynamic request arrivals showing EQSPEC maintains predictable tail latency suitable for online serving, while EXSPEC achieves higher throughput at the cost of increased tail latency, demonstrating why we provide both algorithms for different deployment scenarios.


### Reviewer ajDR
Concerns:
* Models (Vicuna, GLM, Qwen3) are outdated/small
* No KV cache management optimizations
* Methods designed only for offline inference with uniform sequence lengths
* (Round 2) Request to add proposed methods to Figure 6 for direct comparison with vLLM/SGLang

Response: **Most concerns stem from misunderstandings or factual errors**. On model selection: Qwen3-8B and GLM-4-9B are 2025 models; Vicuna-7B is the standard benchmark (EAGLE, DSD, SpecBench); we added larger Qwen3-14B/32B experiments. On KV cache: our work addresses KV cache realignment based on sequence positions within a batch, which is orthogonal to underlying storage mechanisms (paged, tree, or standard transformers cache). Prior methods skip this realignment and produce gibberish. On offline/uniform lengths: EQSPEC supports both online and offline scenarios; SpecBench has 54× length diversity (25-1351 tokens); we provided multi-turn latency experiments under heterogeneous workloads. On Round 2: **direct comparison is infeasible** due to architectural differences between transformers and vllm/sglang. Reviewer acknowledged some concerns addressed and indicated willingness to increase score if Round 2 concern is solved.


### Reviewer RaXj
Concerns:
* "Identical output" claim only valid for temperature=0
* BSP/DSD fail at batch size=1, seemingly contradicting the "batch size >1" claim
* (round 1 & 2) Root causes of existing systems' failures not clearly specified
* (round 2) Figure 5(a) "No-Ragged-Scaling" baseline interpretation confusing
* (round 3) Draft models are weak (<1B); suggested testing larger model e.g., Qwen 2.5 or LLama
* (round 3) Why is exact match ~90% instead of 100% if method guarantees identity?

Response: **Most concerns stem from presentation clarity, which we have addressed thoroughly.** We provided code examples and a literature taxonomy table. We clarified that Figure 5(a) shows batch scaling efficiency normalized to each method's own BS=1, and acknowledged the temperature=0 clarification. For Round 2 & 3: our method already achieves positive speedup on well-established pairs, demonstrating that batch speculation successfully amplifies per-sequence gains. Equation 1 characterizes the relationship between draft model quality, acceptance rate, and speedup. Exploring draft models that better balance inference cost and acceptance rate is valuable future work. For the ~95% exact match, we cited recent work showing floating-point non-determinism causes output variance even under greedy decoding, which is orthogonal to our contribution. The reviewer recognized our contribution and indicated willingness to raise score to leaning acceptance if concerns are solved.

**Current Status**: One accept (8), two conditional increases pending. Both conditional reviewers' Round 2 & 3 concerns have now been addressed in final responses.

---

> ### Author Response · Authors · 2025-12-03
> **Revised Manuscript Uploaded**
>
> We have uploaded a revised manuscript addressing reviewer feedback; key changes are summarized below.
>
> ### Main Paper Revisions
> - **Clarified correctness criterion (Section 2):** Added a paragraph formalizing that for greedy decoding (temperature=0), speculative decoding must yield identical outputs, while for temperature>0, it must preserve the target model’s output distribution. This losslessness defines authentic batch speculative decoding.
> - **Revised Figure 5(a) caption:** Clarified that “No-Ragged-Scaling” represents measured autoregressive throughput normalized to its own BS=1 baseline, not a theoretical upper bound. Each method is normalized independently to isolate batch scaling efficiency.
> - **Enhanced Equation 1 discussion:** Strengthened the connection between our batch-aware speedup model and experimental findings, providing clearer intuition for why alignment overhead eventually overwhelms parallelism gains.
> - **Added latency-throughput analysis (Section 4.4):** Included online serving experiments measuring P50/P90/P99 latencies, demonstrating EQSPEC’s predictable tail latency versus EXSPEC’s throughput optimization.
> ### Appendix Additions
> - **Relocated production system comparison:** Moved vLLM/SGLang scaling figure to appendix due to architectural differences complicating direct comparison.
> - **Added discussion section:** Synthesized key findings on how the ragged tensor problem breaks the multiplicative relationship between batch parallelism and per-sequence speculation. Clarified that batch speculation amplifies per-sequence gains but cannot create them.

---

### Meta-Review · Area_Chair_VBDa · 2026-01-10

**Summary:**

The paper studies the ragged tensor problem in batched speculative decoding (SD), where the accepted draft lengths are different for different queries in a batch, which causes misalignment for the verification phase. The paper proposes algorithms to mitigate this issue and shows that it can help improve decoding inconsistencies in existing batched SD implementations.

The reviewers raised a few concerns that I feel still stand after the rebuttal phase: (1) insufficient separation of causes for the output inconsistency. The main baselines that the paper is comparing to are not consistent for reasons beyond just misalignment due to batching; (2) lack of evaluation of more advanced LLM serving systems (vLLM, SGLang). While the paper shows decoding inconsistencies in these systems, it is unclear whether the proposed fix would mitigate the problem. In fact, vLLM achieves decent matching score on Vicuna models, which shows that the proposed reason may not be necessary for mitigating such inconsistency.

As the proposed methods introduce large overhead to fix the inconsistency, I think the paper would need more extensive study on the effect of such misalignment to support their correctness-first approach.

Additionally, it seems the straightforward fix of writing a customized kernel to fix the inconsistency in position encoding is not sufficiently discussed in the paper. The paper would need more dicussion on why this is not a viable approach.

Given these, I would recommend a rejection on the paper. I think the studied ragged tensor problem is an interesting finding to the community. The paper would benefit from more study on the effect of this issue and I recommend the authors to submit a revised draft to a future venue.

**Reviewer Concerns:**

See discussion in the meta-review.

**Reviewer Scores:**

I think reviewer RaXj and reviewer ajDR might have a chance of increasing their score, but these would not affect my recommendation on this paper.

---

### Decision · Program_Chairs · 2026-01-26

Reject